# Synthesis of New 2,3-Dihydroindole Derivatives and Evaluation of Their Melatonin Receptor Binding Affinity

**DOI:** 10.3390/molecules27217462

**Published:** 2022-11-02

**Authors:** Maria S. Volkova, Alexander M. Efremov, Elena N. Bezsonova, Michael D. Tsymliakov, Anita I. Maksutova, Maria A. Salykina, Sergey E. Sosonyuk, Elena F. Shevtsova, Natalia A. Lozinskaya

**Affiliations:** 1Department of Chemistry, Lomonosov Moscow State University, 119991 Moscow, Russia; 2Institute of Physiologically Active Compounds at Federal Research Center of Problems of Chemical Physics and Medicinal Chemistry, Russian Academy of Sciences (IPAC RAS), 142432 Moscow, Russia

**Keywords:** 2,3-dihydroindole, 2-chloromelatonin, chemoselective reduction, melatonin, nitrile reduction, 2-oxindole reduction

## Abstract

2,3-Dihydroindoles are promising agents for the synthesis of new compounds with neuroprotective and antioxidant properties. Usually, these compounds are obtained by direct reduction of the corresponding indoles containing acceptor groups in the indole ring for its activation. In this work, we propose a synthetic strategy to obtain new 2,3-dihydroindole derivatives from the corresponding polyfunctional 2-oxindoles. Three methods were proposed for reduction of functional groups in the 2-oxindole and 2-chloroindole molecules using various boron hydrides. The possibility of chemoselective reduction of the nitrile group in the presence of an amide was shown. The proposed synthetic strategy can be used, for example, for the synthesis of new analogs of the endogenous hormone melatonin and other compounds with neuroprotective properties.

## 1. Introduction

Melatonin (3-(2-(acetylamino)ethyl)-5-methoxyindole) is a neurohormone playing a central role in the regulation of circadian rhythms in mammals, including humans. In addition, melatonin has an effect on the activities of immune, cardiovascular, and reproductive systems [1,2,3]. Melatonin and its analogues exhibit antidepressant, antioxidant, neuroprotective, hypotensive and anticancer activities [4,5,6,7,8,9,10,11]. The melatonin-based bivalent ligands demonstrate improved ocular hypotensive [12], antioxidant [13] and neuroprotective [14] properties.

Known to date high-affinity ligands for melatonin receptors are diverse in their structures [15]. Despite the fact that many useful ligands contain bioisosteric replacements of the indole ring by an aromatic or heteroaromatic ring, the scope of melatonin-like modifications of the indole ring is far from being exhausted. We assumed that 2,3-dihydroindoles are a class of compounds that could be easily modulated to provide different receptor subtype selectivity and intrinsic activity profiles. The structural requirements and pharmacophore groups necessary for the binding to the MT_1_/MT_2_ and MT_3_ melatonin receptors are shown in Figure 1A,B.

Group **I**, which is either a methoxy group or its homologous (cyclic) analogue, is necessary for the compound to demonstrate agonistic activity toward the MT_1_/MT_2_ melatonin receptor subtypes, whereas the presence of this group is not essential for antagonists, for example agomelatin. The presence of amide group **II** and spacer **III**, which is exactly equal to two carbon atoms, is also necessary for binding affinity [16]. Structural requirements for low-affinity MT_3_ receptor ligands are more tentative. Since the MT_3_ receptor is apparently the enzyme quinone reductase 2 with planar FAD cofactor [17], it might be expected that a “good” ligand would contain a planar aromatic ring. It is also known that the presence of carbamoyl group **IV** substantially improves binding to the MT_3_ receptor.

Before we present a general and robust approach to synthesize 2-oxoindolylacetonitriles [18,19,20]. Herein, we report a modification of this method that aims to create a sp^3^-carbon atom in position 3 of the indole ring and introduce an additional substituent R^3^ (see Figure 1C). The escape of the acetamide side chain from the plane of the indole moiety can increase activity of the compound toward the MT_1_/MT_2_ melatonin receptors [21]. The new compounds with different substituents in positions 2 and 5 of the indole ring were synthesized using this novel approach (see Figure 1C and Figure 1).

## 2. Results and Discussion

### 2.1. Chemistry

The key step of the proposed approach to the synthesis of various melatonin receptor ligands involves the Knoevenagel condensation of isatins with cyanoacetic acid or its esters (Figure 1):

A series of isatins **1** was synthesized from aniline precursors by the Sandmeyer method and was additionally modified at the indole nitrogen by alkylation according to the described procedure [22] (Figure 2). The condensation of isatins **1e**–**k** and **1m**–**n** with cyanoacetic acid was performed in the presence of triethylamine according to the procedure described for isatins **1a**–**c** and **1l** [23].

We proposed two synthetic routes to various (2-oxoindolin-3-yl)acetonitriles, which both consist of reduction of the double bond and decarboxylation of Knoevenagel condensation products **2a**–**j** (procedures **A** and **B**, see Figure 3).

The approach (**A**) involves reduction of the double bond followed by decarboxylation of the resulting acid. The palladium-catalyzed hydrogenation of the double bond of unsubstituted (2-oxoindolin-3-yl)cyanoacetic acid was described in the literature [24]; however, partial reduction of the nitrile group occurs as a side process. Hence, we developed and optimized the reduction of the double bond in compounds **2a**–**2j** using the Zn/aq. HCl system. Reduction products were subjected to the decarboxylation without characterization and additional purification. Nitriles **4** were obtained with good yields (Figure 3, Table 1). The alternative approach **B** (see stages **b**–**c**, Figure 3) involves decarboxylation of **2** in pyridine to obtain compounds **3** followed by reduction of the double bond. Compounds **4c**–**4j** were synthesized using this procedure, which was described in the literature for the synthesis of compounds **4a–c [18,25]**. The overall yields after two steps are given in Table 1.

In a molecule of 3-alkyl-substituted 2,3-dihydromelatonins acetamide, the side chain is shifted from the plane of the indole moiety and is locked into this conformation. To investigate the influence of these conformational features on affinity to melatonin receptors, we synthesized a few novel 3-substituted nitriles by selective alkylation of nitriles **4a**–**c** using various alkyl halides.

To avoid undesired *N*-alkylation, the Boc-protecting group was chosen. We found that either *N*-Boc-acylation products or 1,3-Boc derivatives could be produced depending on the reaction conditions (Figure 4).

Similarly, alkylation of **4b** with MeI in the presence of DMAP led to a mixture of mono- and dialkylation products (see Materials and methods).

*N*-Boc-and *N*-alkyl-substituted nitriles were alkylated at position 3 with various alkylating agents in the presence of sodium hydride (see Figure 3, Table 2).

The obtained (2-oxoindolin-3-ylidene)acetonitriles **3** were also used as precursors of conformationally restricted spiro derivatives. Initially, we intended to use the Corey–Chaykovsky reaction for the synthesis of spirocyclopropane 2-oxindoles. We performed the reaction of esters **2** with trimethylsulfoxonium iodide in the presence of sodium hydride according to standard procedures for cyclopropanation of the double bond bearing two electron-withdrawing substituents. However, the reaction of methyl (2-oxoindolin-3-ylidene)cyanate with sulfur ylide produced an inseparable mixture of compounds. Hence, we tested the method based on the [3 + 2]-cycloaddition of diazomethane to compounds **3** in the same way as we previously described for compounds **6a**, **b** [19]. The synthesis was carried out in the presence of a 20-fold excess of diazomethane without any catalyst. The resulting pyrazolines were immediately subjected to thermal decomposition with no additional purification. Spirocyclopropane derivatives **6a**, **b**, **e**, **g**, **j** were synthesized in high yields (see Table 3). The low yield in case of nitriles **6c**, **d**, **i** can be attributed to side reactions, in particular, to the partial consumption of diazomethane in the methylation of nitrogen in position 1.

The side chain in position 3 of the indole molecule containing the acetamide group plays a key role in the affinity of the compounds for the MT_1_ and MT_2_ melatonin receptors. Hence, an important part of the present study was to develop different methods for reduction of the nitrile group and acetylation of the resulting amines. Previously, we have shown that the reduction of CN group of (2-oxoindolin-3-yl)acetonitrile using hydrogenation with PtO_2_ as a catalyst in the presence of acetic anhydride is a decent approach to a new melatonin derivatives synthesis [19,20]. A new 2-oxoindole-based melatonin analog **7a** was synthesized in the present study using this method (Figure 5).

Selective nitrile reduction in the presence of cyclic amide was also achieved using NaBH_4_ in methanol in the presence of catalytic amounts of anhydrous NiCl_2_. We have previously shown that the addition of acetic anhydride to the reduction mixture could obtain 2-chloromelatonin **7d** [26] with good yield (Figure 6). In this work, we demonstrated that the same reduction followed by *one-pot* acylation can be used in the case of 2-oxindole derivatives and other anhydrides (Figure 6).

The observed lower yields compared to catalytic hydrogenation on PtO_2_ for 2-oxoderivatives **7a** can be explained by loss catalytic activity of nickel boride due to its coordination on amide group.

2,3-Dihydro derivatives of melatonin were synthesized by reduction of nitriles **4** by in situ generated BH_3_ using an excess of sodium borohydride in the presence of iodine in dry tetrahydrofuran. However, for 3-monosubstituted oxindoles, the formation of 2,3-dihydromelatonin (**8**) under these reaction conditions is accompanied by the partial aromatization of the indole ring (**9**). The reduction of nitriles **3** containing a double bond under these conditions also leads to products **8** and **9** (Figure 7).

The spontaneous aromatization cannot occur in the case of reduction of (indolin-3-yl)acetonitriles containing substituents in positions 1 and 3 of the indole ring (Figure 8). The reduction of nitriles **5** and spironitriles **6** will afford new stable 2,3-dihydroindoles **8**,**10** (Table 4).

Under these conditions, Boc-containing compounds completely reduce all functional groups, including Boc: the tert-butoxycarbonyl group was reduced to methyl (Figure 9).

Despite stability in the solid phase, the obtained 2,3-dihydroindoles are sensitive to oxidation in solution. Thus, the oxidation in the NMR tube occurred both in spiro and 3H-containing indolines, but led to different types of products. For compound **8m**, aromatization of the indole ring occurred, while 3,3-disubstituted indoline **10a** was oxidized to 2-oxindole (Figure 10). The compounds **8i**,**l**,**g** were also partially converted to aromatic indoles **9i**,**l**,**g** during purification by column chromatography.

### 2.2. Melatonin Receptor Binding Activity

The newly synthesized indole derivatives were evaluated for their binding affinity and intrinsic activity at human MT_1_ and MT_2_ receptors stably transfected in Chinese hamster ovary (CHO) cells using 2-[^125^I]iodomelatonin as a radioligand, and the results are shown in Table 5 and Table 6.

First, compounds **7a**, **7d** containing heteroatoms in position 2 were evaluated for MT binding assay using melatonin and selective MT_2_ receptor antagonist 4-P-PDOT as reference (Table 5 and Appendix A). The presence of 2-oxindole ring dramatically decreased MT_1_/MT_2_ receptor binding affinity for melatonin derivatives while 2-chloromelatonin **7d** was more active than melatonin with respect to both types of MT receptors.

The same tendency was observed in the case of 2,3-dihydroindoles: their binding affinity to both types of MT receptors was sufficiently lower than the activity of melatonin (Table 6).

## 3. Materials and Methods

### 3.1. Chemistry

All solvents were used as received without further purification. The reactions were monitored by thin layer chromatography (TLC) carried out on Merck TLC silica gel plates (60 F254), using UV light for visualization and basic aqueous potassium permanganate or iodine fumes as developing agent. Flash column chromatography purifications were carried out using silica gel 60 (particle size 0.040–0.060 mm).

^1^H and ^13^C NMR spectra were recorded at 298 K on Bruker Avance 300 spectrometer with operating frequency of 400.13 and 100.6 MHz, respectively, and calibrated using residual CHCl_3_ (δH = 7.26 ppm) and CDCl_3_ (δC = 77.16 ppm) or DMSO-d_5_ (δH = 2.50 ppm) and DMSO-d_6_ (δC = 39.52 ppm) as internal references. NMR data were presented as follows: chemical shift (δ ppm), multiplicity (s = singlet, d = doublet, dd = doublet of doublet, t = triplet, q = quartet, m = multiplet, br. = broad), coupling constant (J) in Hertz (Hz), integration. High-resolution mass spectra (HRMS) were measured on a Thermo Scientific LTQ Orbitrap instrument using nanoelectrospray ionization (nano-ESI). Isatin **1a**, 5-methoxyisatin **1b**, 5-bromoisatin **1c** and 6,7-dimethylisatin **1d** were purchased from Merck. The following isatin derivatives were obtained by *N*-alkylation using previously described procedure [28]: *N*-methylisatin **1e**, *N*-benzylisatin **1f**, *N*-methyl-5-methoxyisatin **1g**, *N*-benzyl-5-methoxyisatin **1h**, *N*-methyl-5-bromoisatin **1i**, *N*-methyl-6,7-dimethylisatin **1j**. The following compounds were obtained as previously described: cyano(2-oxoindolin-3-ylidene)acetic acid **2a** [25], cyano(2-oxo-5-methoxy-indolin-3-ylidene)acetic acid **2b** [24], cyano(2-oxo-5-bromoindolin-3-ylidene)acetic acid **2c** [23], 2-(2-oxo-2,3-dihydro-*1H*-indol-3-ylidene)acetonitrile **3a**, [25] 2-(1-methyl-2-oxo-2,3-dihydro-*1H*-indol-3-ylidene)acetonitrile **3e** [24], 2-(5-methoxy-1-methyl-2-oxo-2,3-dihydro-*1H*-indol-3-ylidene)acetonitrile **3g** [29], 2’-oxo-1’,2’-dihydrospiro[cyclopropane-1,3′-indol]-3-carbonitrile (**6a**) [19], 5′-methoxy-2’-oxo-1’,2’-dihydrospiro[cyclopropane-1,3′-indol]-3-carbonitrile (**6b**) [19], 2-chloromelatonin **7d** [26].

#### 3.1.1. General Procedure for Synthesis of cyano(2-oxoindolin-3-ylidene)acetic Acids (**2**)

Isatin **1** (1 eq.) was dissolved in warm abs. dioxane (ca. 3 mL dioxane per 1 g of isatin) and mixture of cyanoacetic acid (1eq) and triethylamine (1.2 eq) in dioxane (ca. 1 mL dioxane per 1 g of acid) was added. Reaction mixture was vigorously stirred for 4–5 h, and then, reaction was terminated by addition of 30 mL conc. HCl. The reaction mixture was stored at room temperature for 1–5 days until precipitate was obtained. The precipitate was filtered and washed with cold water. All obtained compounds were sufficiently pure (according to ^1^H NMR) and could be used in further synthesis without additional purification. The following compounds were obtained according to this procedure:
Cyano(6,7-dimethyl-2-oxo-indolin-3-ylidene)acetic acid (**2d**)

From 8.00 g (0.046 mol) of 6,7-dimethylisatin (**1d**), 3.90 g (0.046 mol) of cyanoacetic acid, 6.70 mL (0.055 mol) of triethylamine in 28 mL of 1,4-dioxane brown powder (5.95 g, yield 54%) was obtained; m.p. = 222–230 °C. NMR ^1^H (DMSO-d_6_): δ 1.52 (s, 3H), 1.67 (s, 3H), 6.26 (d, J = 7.6 Hz, 1H), 7.27 (d, J = 7.5, 1H), 10.8 (s, 1H). NMR ^13^C (DMSO-d_6_): δ 13.46, 17.91, 25.44, 50.79, 119.83, 122.07, 125.44, 128.42, 130.01, 147.88, 151.41, 166.22, 173.57. Elemental analysis found (%): C 64.42, H 4.18, N 11.51, calculated for C_13_H_10_N_2_O_3_ (%): C 64.46, H 4.16, N 11.56.

Cyano(*N*-methyl-2-oxoindolin-3-ylidene)acetic acid (**2e**)

From 6.87 g (0.043 mol) of *N*-methylisatin (**1e**), 3.62 g (0.043 mol) of cyanoacetic acid, 5.68 mL (0.052 mol) of triethylamine in 35 mL of 1,4-dioxane dark-cherry solid was obtained (7.40 g, 75% yield); m.p. = 177–178 °C. NMR ^1^H (DMSO-d_6_): δ 3.13 (s, 3H), 6.85 (d, J = 7.8, 1H), 7.05 (t, J = 7.6, 1H), 7.41 (t, J = 7.6, 1H), 7.95 (d, J = 7.8, 1H). NMR ^13^C (DMSO-d_6_): δ 26.32, 108.45, 109.59, 114.63 (CN), 122.97, 123.39, 124.71, 134.63, 138.79, 145.67, 161.79, 164.05. Elemental analysis found (%): C 63.21, H 3.68, N 12.11, calculated for C_12_H_8_N_2_O_3_ (%): C 63.16, H 3.53, N 12.28.

Cyano(*N*-benzyl-2-oxoindolin-3-ylidene)acetic acid (**2f**)

From 7.56 g (0.032 mol) of *N*-benzylisatin (**1f**), 2.9 g (0.032 mol) of cyanoacetic acid, 7.70 mL (0.056 mol) of triethylamine and 50 mL in 1,4-dioxane brown powder was obtained (7.98 g, 82% yield); m.p. = 169–173 °C. NMR ^1^H (CDCl_3_): δ 4.55 ** (s, 2H), 4.58 * (s, 2H), 6.45 ** (d, J = 8.1, 1H), 6.42 * (d, J = 8.1, 1H), 6.64 * (t, J = 7.6, 1H), 6.75 ** (t, J = 7.8, 1H), 6.95 (m, 5H), 7.02 ** (t, J = 7.8, 1H), 7.72 ** (d, J = 7.6, 1H), 7.91 * (d, J = 7.8, 1H). **—major isomer, *—minor isomer. Ratio of isomers was 4:1. NMR ^13^C (CDCl_3_): δ 43.69, 108.49, 110.28, 114.61, 114.78, 123.46, 124.67, 134.46, 127.31, 128.93, 129.4, 138.54, 134.87, 144.8, 161.85, 164.35. Elemental analysis found (%): C 70.07, H 3.92, N 9.22, calculated for C_18_H_12_N_2_O_3_ (%): C 71.05, H 3.97, N 9.21.

Cyano(*N*-methyl-2-oxo-5-methoxy-indolin-3-ylidene)acetic acid (**2g**)

From 3.32 g (0.0173 mol) of *N*-methyl-5-methoxyisatin (**1g**), 1.51 g (0.0178 mol) of cyanoacetic acid, 3.0 mL (0.0238 mol) of triethylamine in 18 mL of 1,4-dioxane dark-violet solid (3.08 g, yield 69%) was obtained; m.p. = 182–183 °C. NMR ^1^H (CDCl_3_): δ major isomer: 3.16 (s, 3H), 3.71 (s, 3H), 6.79 (d, J = 8.8, 1H), 6.99 (dd, J = 8.6, J = 2.3, 1H), 7.82 (d, J = 2.8, 1H), δ minor isomer: 3.05 (s, 3H), 3.64 (s, 3H), 6.61 (d, J = 8.3, 1H), 6.89 (dd, J = 8.6, J = 2.5, 1H), 7.78 (d, J = 2.8, 1H), Ratio of isomers was 10:1. NMR ^13^C (CDCl_3_): δ major isomer 26.48, 55.40, 110.67, 112.09, 114.21, 116.21, 119.13, 120.65, 137.95, 141.96, 150.60, 156.74, 159.94. Elemental analysis found (%): C 60.57, H 3.86, N 10.22, calculated for C_13_H_10_N_2_O_4_ (%): C 60.47, H 3.90, N 10.85.

Cyano(*N*-benzyl-2-oxo-5-methoxyindolin-3-ylidene)acetic acid (**2h**)

From 7.42 g (0.028 mol) of *N*-benzyl-5-methoxyisatin (**1h**), 2.47 g (0.029 mol) of cyanoacetic acid, 4.20 mL (0.03 mol) of triethylamine in 45 mL of 1,4-dioxane violet powder was obtained (6.42 g, yield 68%); m.p. = 152–153 °C. NMR ^1^H (DMSO-d_6_): δ 3.69 ** (s, 3H), 3.65 * (s, 3H), 4.79 ** (s, 2H), 4.81 * (s, 2H), 6.63 ** (d, J = 8.6, 1H), 6.6 * (d, J = 8.59, 1H), 6.85 ** (dd, J = 8.6, J = 2.5, 1H), 6.84 * (dd, J = 8.6, J = 2.5, 1H); 7.21 (m, 5H); 7.54 ** (d, J = 2.5, 1H); 7.81 * (d, J = 2.5, 1H). **—major isomer, *—minor isomer. Ratio of isomers was 4:1. NMR ^13^C (DMSO-d_6_): δ 43.71, 55.83, 108.75, 111.10, 114.61, 115.33, 119.72, 120.63, 127.35, 128.45, 129.97, 135.02, 138.53, 139.15, 156.06, 161.73, 164.08. Elemental analysis found (%): C 68.33, H 4.18, N 8.32, calculated for C_18_H_12_N_2_O_3_ (%): C 68.26, H 4.22, N 8.38.

Cyano(*N*-methyl-2-oxo-5-bromoindolin-3-ylidene)acetic acid (**2i**)

From 5.13 g (0.021 mol) of *N*-methyl-5-bromoisatin (**1i**), 1.82 g (0.021 mol) of cyanoacetic acid, 2.80 mL (0.027 mol) of triethylamine in 17 mL of 1,4-dioxane violet powder was obtained (5.90 g, yield 89%); m.p. = 137–138 °C. NMR ^1^H (CDCl_3_): δ 3.14 (s, 3H); 6.95 (d, J = 8.4, 1H), 7.61 (dd, J = 8.4, J = 1.9, 1H), 8.28 (d, J = 1.9, 1H). NMR ^13^C (CDCl_3_): δ 21.48, 105.49, 109.20, 115.26, 122.47, 127.07, 132.57, 139.57, 140.36, 156.42, 158.11, 166.82. Elemental analysis found (%): C 47.00, H 3.83, N 9.17, calculated for C_12_H_7_BrN_2_O_3_ (%): C 46.93, H 2.30, N 9.12.

Cyano(1,6,7-trimethyl-2-oxoindolin-3-ylidene)acetic acid (**2j**)

From 5.00 g (0.027 mol) of *N*-methyl-6,7-dimethylisatin (**1j**), 2.30 g (0.027 mol) of cyanoacetic acid, 3.50 mL (0.028 mol) of triethylamine in 17 mL of 1,4-dioxane light brown solid (3.76 g, yield 59%) was obtained; m.p. = 177–180 °C. NMR ^1^H (CDCl_3_): δ 3.21 (s, 3H), 2.34 (s, 3H), 2.55 (s, 3H), 3.59 (s, 3H), 7.12 (s, 1H), 8.17 (s, 1H). NMR ^13^C (CDCl_3_): δ 14.01, 21.51, 30.57, 95.23, 116.48, 118.22, 119.38, 119.89, 122.37, 125.01, 143.08, 143.79, 145.08, 168.75. Elemental analysis found (%): C 65.53, H 4.80, N 10.85, calculated for C_14_H_12_N_2_O_3_ (%): C 65.62, H 4.72, N 10.93.

#### 3.1.2. General Procedure for Synthesis (2-oxoindolin-3-ylidene)acetonitriles (**3**)

Solution of compound **2** in pyridine was heated using water bath for 2 h. Then, reaction mixture was cooled, and acetic acid was added. Precipitate was collected, washed with cold water and dried. According to this procedure, the following compounds were obtained:
2-(5-Methoxy-2-oxo-2,3-dihydro-*1H*-indol-3-ylidene)acetonitrile (**3b**)

1.00 g (4.1 mmol) of cyano(5-methoxy-2-oxo-indolin-3-ylidene)acetic acid (**2b)** in 2 mL of pyridine was heated for 2 h before the 10 mL of acetic acid was added. Dark red precipitate (0.52 g, yield 64%) was obtained; m.p. = 290–292 °C. NMR ^1^H (DMSO-d_6_): δ 3.73 (s, 3H), 6.47 (s, 1H), 6.79 (d, J = 8.6, 1H), 7.03 (dd, J = 8.6, J = 2.5, 1H), 7.35 (d, J = 2.3, 1H), 10.65 (s, 1H). IR, cm^−1^: 1600 (Ar), 1620 (C=CH-CN), 1730 (NHC(O)), 2220 (CN), 3150–3300 (NH). Elemental analysis found (%): C 65.95, H 4.03, N 13.92; calculated for C_11_H_8_N_2_O_2_ (%): C 66.00, H 4.03, N 13.99.

2-(5-Bromo-2-oxo-2,3-dihydro-*1H*-indol-3-ylidene)acetonitrile (**3c**)

5.00 g (0.017 mol) of cyano(5-bromo-2-oxoindolin-3-ylidene)acetic acid **2c** in 20 mL of pyridine was heated for 2 h, and then 50 mL of acetic acid was added. Dark red precipitate was obtained, 2.84 g, yield 66%; m.p. = 238–239 °C. NMR ^1^H (CDCl_3_): δ 6.49 ** (s, 1H), 6.59 * (s, 1H), 6.79 ** (d, J = 8.2, 1H), 6.85 * (d, J = 8.2, 1H), 7.50 ** (d, J = 8.2, 1H), 7.57 * (d, J = 8.2, 1H), 7.84 * (s, 1H), 7.93 ** (s, 1H), 10.84 ** (s, 1H), 10.99 * (s, 1H). **—major isomer, *—minor isomer. Ratio of isomers was 10:1. NMR ^13^C (CDCl_3_): δ 99.91, 113.36, 114.04 (CN), 116.92, 121.73, 126.48, 136.45, 143.02, 144.28, 165.92 (C=O).

2-(6,7-Dimethyl-2-oxo-2,3-dihydro-*1H*-indol-3-ylidene)acetonitrile (**3d**)

5.85 g (0.029 mol) of cyano(6,7-dimethyl-2-oxo-indolin-3-ylidene)acetic acid **2d** and 12 mL of pyridine were heated for 2 h, and then 70 mL of acetic acid was added. Resulting 3.11 g of dark red precipitate had been obtained, yield 63%; m.p. = 207 °C. NMR ^1^H (CDCl_3_): δ 1.71 (s, 3H), 1.85 (s, 3H), 5.89 (s, 1H), 6.40 (d, J = 7.6, 1H), 7.23 (d, J = 7.6, 1H), 10.1 (s, 1H). NMR ^13^C (CDCl_3_): δ 12.85, 20.23, 94.18, 116.34 (CN), 117.13, 119.04, 121.57, 123.55, 143.35, 144.93, 167.96 (C^2^ = O). Elemental analysis found (%): C 72.65, H 5.04, N 14.10, calculated for C_12_H_10_N_2_O (%): C 72.71, H 5.08, N 14.13.

2-(1-Benzyl-2-oxo-2,3-dihydro-*1H*-indol-3-ylidene)acetonitrile (**3f**)

7.80 g (0.026 mol) of cyano(*N*-benzyl-2-oxoindolin-3-ylidene)acetic acid **2f** in 20 mL of pyridine was heated for 2 h, and then 90 mL of acetic acid was added. Dark red precipitate was obtained (4.50 g, yield 68%); m.p. = 155 °C. NMR ^1^H (CDCl_3_): δ 4.93 (s, 2H), 6.39 (s, 1H), 6.75 (d, J = 7.8, 1H), 7.1 (t, J = 7.6, 1H), 7.33 (m, 6H), 8.09 (d, J = 7.6, 1H). NMR ^13^C (CDCl_3_): δ 43.55 (CH_2_), 98.85 (=C(CN)), 110.34 (C^7^), 116.73 (CN), 119.24 (C^5^), 123.25 (C^4^), 124.47 (C^3a^), 127.47, 127.92, 128.96, 133.88 (C_Bn_), 135.97 (C^6^), 143.08 (C^3^), 145.3 (C^3b^), 165.17 (C^2^=O). IR, cm^−1^: 2220 (-CN), 1715 (CO), 1610 (C=C). Elemental analysis found (%): C 78.50, H 4.64, N 10.79, calculated for C_17_H_12_N_2_O (%): C 78.44, H 4.65, N 10.76.

2-(1-Benzyl-5-methoxy-2-oxo-2,3-dihydro-*1H*-indol-3-ylidene)acetonitrile (**3h**)

2.90 g (0.0086 mol) of cyano(*N*-benzyl-2-oxo-5-methoxyindolin-3-ylidene)acetic acid **2h** in 6 mL of pyridine was heated for 2 h, and then 27 mL of acetic acid was added. Dark red precipitate was obtained, 1.71 g, yield 91%; m.p. = 126–127 °C. NMR ^1^H (DMSO-d_6_): δ 3.72 (s, 3H), 4.88 (s, 2H), 6.67 (s, 1H), 6.9 (d, J = 8.8, 1H), 7.01 (dd, J = 2.5, J = 8.6, 1H), 7.31 (m, 5H), 7.44 (d, J = 2.5, 1H). NMR ^13^C (DMSO-d_6_): δ 43.93 (CH_2_), 55.85 (MeO), 97.82 (=**C**(CN)), 110.57 (C^4^), 116.03 (C^7^), 119.36 (CN), 119.88 (C^6^), 124.88 (C^3a^), 127.24, 127.91, 128.89, 134.99 (C_Bn_), 138.59 (C^3^), 143.78 (C^3b^), 154.33 (C^5^), 165.22 (C^2^=O). IR, cm^−1^: 2230 (-CN), 1710 (CONH), 1620 (C=C). Elemental analysis found (%): C 74.44, H 4.81, N 9.66, calculated for C_18_H_14_N_2_O_2_ (%): C 74.47, H 4.86, N 9.65.

2-(5-Bromo-1-methyl-2-oxo-2,3-dihydro-*1H*-indol-3-ylidene)acetonitrile (**3i**)

1.04 g (0.0049 mol) of cyano(*N*-methyl-2-oxo-5-bromoindolin-3-ylidene)acetic acid **2i** and 4 mL of pyridine were heated for 2 h, then 20 mL of acetic acid was added. Resulting red precipitate was obtained, single isomer, yield 0.43 g, 54%; m.p. = 195–196 °C. NMR ^1^H (CDCl_3_): δ 3.23 (s, 3H), 6.38 (s, 1H), 6.74 (d, J = 8.3, 1H), 7.57 (dd, J = 1.5, J = 8.3, 1H), 8.16 (d, J = 1.5, 1H). NMR ^13^C (CDCl_3_): δ 26.42, 99.10, 110.32, 115.56, 120.67, 125.03, 127.66, 136.27, 142.33, 144.54, 164.70. Elemental analysis found (%): C 50.34, H 2.71, N 10.66, calculated for C_11_H_7_BrN_2_O (%): C 50.22, H 2.68, N 10.65.

2-(1,6,7-Trimethyl-2-oxo-2,3-dihydro-*1H*-indol-3-ylidene)acetonitrile (**3j**)

1.00 g (0.015 mol) of cyano(1,6,7-trimethyl-2-oxoindolin-3-ylidene)acetic acid **2j** and 4 mL of pyridine were heated, and then 20 mL of acetic acid was added. Resulting 0.45 g of dark violet precipitate had been obtained, mixture of isomers Z:E = 1:10, yield 52%; m.p. = 187 °C. Major isomer: NMR ^1^H (CDCl_3_): 2.34 (s, 3H), 2.43 (s, 3H), 3.50 (s, 3H), 6.17 (s, 1H), 6.90 (d, J = 7.7, 1H), 7.80 (d, J = 7.7, 1H). NMR ^13^C (CDCl_3_): 13.96, 21.46, 30.52, 95.18, 116.43 (CN), 118.17, 119.84, 122.32, 124.96, 143.03, 143.74, 145.02, 166.70 (C^2^ = O). Minor isomer: NMR ^1^H (CDCl_3_): 2.34 (s, 3H), 2.43 (s, 3H), 3.47 (s, 3H), 5.94 (s, 1H), 6.85 (d, J = 7.6, 1H), 7.16 (d, J = 7.7, 1H). Elemental analysis found (%): C 73.57, H 5.66, N 13.19, calculated for C_13_H_12_N_2_O (%): C 73.56, H 5.70, N 13.20.

#### 3.1.3. General Procedure for Synthesis of (2-oxo-2,3-dihydro-1H-indol-3-yl)acetonitriles (**4**)

Method A:

To a solution of **2** in mixture of ethyl acetate and 3N hydrochloric acid, the excess of zinc dust was added, and the reaction mixture was vigorously stirred for 0.5 h. The color of the reaction mixture turned from dark red to pale yellow. Then, organic phase was separated, washed with cold distilled water and dried with Na_2_SO_4_. The solution was concentrated in vacuum. The obtained cyano(2-oxo-2,3-dihydro-*1H*-indol-3-yl)acetic acid was used in the decarboxylation step without additional purification. The residue was dissolved in 2-ethoxyethanol and stirred with reflux for 2–2.5 h. Reaction mixture was concentrated and purified by filtration through silica gel pad with ethyl acetate as eluent. The following compounds were obtained according to this procedure:
2-(2-Oxo-2,3-dihydro-*1H*-indol-3-yl)acetonitrile (**4a**) [25]

First, 10.25 g (0.048 mol) of cyano(2-oxo-indolin-3-ylidene)acetic acid **2a**, 40 mL of ethyl acetate, 20 mL of 3N HCl, and 4.80 g (0.08 mol) of zinc dust were involved in the reaction. The organic phase was separated and concentrated. The dry residue was dissolved in 30 mL of 2-ethoxyethanol, refluxed for 2 h, concentrated and filtered through silica gel pad using ethyl acetate as eluent. As a result, 5.68 g of beige powder was obtained, yield 69%; m.p. = 155–160 °C (m.p.^lit.^ 160–161 °C [25]). NMR ^1^H (CDCl_3_): 2.61 (dd, J = 16.8, J = 8.2, 1H), 2.88 (dd, J = 16.8, J = 8.2, 1H), 3.44 (m, 1H), 6.74 (d, J = 6.5, 1H), 6.84 (t, J = 5.6, 1H), 7.05 (t, J = 6.1, 1H), 7.22 (d, J = 6.3, 1H), 10.03 (s, 1H, NH).

2-(5-Methoxy-2-oxo-2,3-dihydro-*1H*-indol-3-yl)acetonitrile (**4b**) [24]

First, 2.54 g (0.01 mol) of cyano(5-methoxy-2-oxo-indolin-3-ylidene)acetic acid **2b**, 40 mL of ethyl acetate, 10 mL of 3N HCl and 1.20 g (0.02 mol) of zinc dust were involved in reaction. The organic phase was separated and concentrated. The dry residue was dissolved in 15 mL of 2-ethoxyethanol, refluxed for 2 h concentrated and filtered through silica gel pad using ethyl acetate as eluent. As a result, 1.09 g of beige solid was obtained, yield 57%; m.p. = 179–180 °C (m.p.^lit^ = 180–181 °C [24]). NMR ^1^H (CDCl_3_): 2.77 (dd, J = 17.0, J = 9.1, 1H), 3.10 (dd, J = 17.0, J = 4.8, 1H), 3.70 (dd, J = 8.6, J = 4.3, 1H), 3.81 (c, 3H), 6.89 (dd, J = 8.6, J = 1.2, 1H), 7.02 (c, 1H), 7.20 (d, J = 8.6, 1H), 8.45 (s, 1H, NH). IR, cm^−1^: 2340 (NH), 2280 (-CN), 1700 (CO). MS-EI, 70 eV, m/z: 202 (25%), 175 (2%), 162 (100%), 176 (100%), 147 (14%), 131 (13%), 119 (18%), 104 (11%), 91 (9%), 77 (14%).

2-(5-Bromo-2-oxo-2,3-dihydro-*1H*-indol-3-yl)acetonitrile (**4c**) [23]

First, 5.22 g (0.018 mol) of cyano(5-bromo-2-oxoindolin-3-ylidene)acetic acid **2c**, 36 mL of ethyl acetate, 18 mL of 3N HCl and 2.50 g (0.04 mol) of zinc dust were involved in the reaction. The organic phase was separated and concentrated. The dry residue was dissolved in 15 mL of 2-ethoxyethanol, refluxed for 2.5 h concentrated and filtered through silica gel pad using ethyl acetate as eluent. As a result, 2.02 g of beige solid was obtained, yield 45%; m.p. = 209–210 °C (m.p.^lit.^ 211 °C [23]). NMR ^1^H (CDCl_3_): 3.11 (dd, 1H, J = 5.8, J = 5.8), 3.26 (dd, 1H, J = 5.9, J = 5.8), 3.88 (d, 1H, J = 5.5), 6.84 (d, 1H, J = 8.6), 7.42 (d, 1H, J = 8.4), 7.60 (s, 1H), 10.74 (s, 1H). NMR ^13^C (CDCl_3_): 17.87, 41.90, 111.94, 113.83, 118.50, 127.67, 130.09, 131.70, 142.63, 176.52. MS-EI (mz, %): 252, 250 (M+, 75%), 212, 210 (100%), 195, 197 (10%), 182, 184 (10%), 143 (5%), 116 (15%), 76 (20%).

2-(1-Methyl-2-oxo-2,3-dihydro-*1H*-indol-3-yl)acetonitrile (**4e**) [24]

First, 7.40 g (0.0324 mol) of cyano(*N*-methyl-2-oxo-indolin-3-ylidene)acetic acid **2e**, 54 mL of ethyl acetate, 27 mL of 3N HCl and 5.40 g (0.076 mol) of zinc dust were involved in reaction. The organic phase was separated and concentrated. The dry residue was dissolved in 30 mL of 2-ethoxyethanol, refluxed for 2 h concentrated and filtered through silica gel pad using ethyl acetate as eluent. As a result, 5.30 g of beige solid was obtained, yield 87%; m.p. = 91 °C (m.p^lit^. 89–90 °C [24], 71–72 °C [30]). NMR ^1^H (CDCl_3_): 2.64 (dd, J = 16.8, J = 8.6, 1H), 3.00 (dd, J = 16.7, J = 4.8, 1H), 3.14 (s, 3H), 3.58 (dd, J = 8.3, J = 4.8, 1H), 6.83 (d, J = 7.8, 1H), 7.05 (t, J = 7.6, 1H), 7.29 (t, J = 7.6, 1H), 7.42 (t, J = 7.3, 1H). IR, cm^−1^: 2230 (-CN), 1725 (NH-CO).

2-(1-Benzyl-2-oxo-2,3-dihydro-*1H*-indol-3-yl)acetonitrile (**4f**) [24]

First, 2.35 g (0.0077 mol) of cyano(*N*-benzyl-2-oxoindolin-3-ylidene)acetic acid **2f**, 15 mL of ethyl acetate, 8 mL of 3N HCl and 1.20 g (0.0177 mol) of zinc dust were involved in reaction. The organic phase was separated and concentrated. The dry residue was dissolved in 20 mL of 2-ethoxyethanol, refluxed for 2 h concentrated and filtered through silica gel pad using ethyl acetate as eluent. As a result, 5.30 g of beige solid was obtained, yield 60%; m.p. = 152–154 °C (m.p.^lit^ 150–151 °C [24]). NMR ^1^H (CDCl_3_): 2.76 (d, J = 15.9, 1H), 2.96 (d, J = 15.9, 1H), 3.10 (dd, J = 16.7, J = 4.8, 2H), 3.68 (t, J = 4.6, 1H), 4.88 (d, J = 10.1, 2H), 6.73 (d, J = 7.8, 1H), 7.03 (t, J = 7.3, 1H), 7.22 (m, 5H), 7.44 (d, J = 7.3, 1H). NMR ^13^C (CDCl_3_): 18.97, 41.40, 43.95, 109.64, 117.14, 123.12, 124.21, 127.77, 128.83, 135.21, 126.65, 127.23, 129.25, 143.25, 162.8. IR, cm^−1^: 2220 (-CN), 1715 (NH-CO).

2-(1-Methyl-5-methoxy-2-oxo-2,3-dihydro-*1H*-indol-3-yl)acetonitrile (**4g**) [29]

First, 2.58 g (0.01 mol) of cyano(*N*-methyl-2-oxo-5-methoxy-indolin-3-ylidene)acetic acid **2g**, 22 mL of ethyl acetate, 11 mL of 3N HCl and 1.80 g (0.0277 mol) of zinc dust were involved in the reaction. The organic phase was separated and concentrated. The dry residue was dissolved in 20 mL of 2-ethoxyethanol and orange solution refluxed for 1.5 h concentrated and filtered through silica gel pad using ethyl acetate as eluent. As a result, 1.519 g of beige solid was obtained, yield 67%; m.p. = 116–118 °C (m.p.^lit.^ 115–116 °C [29,30]). NMR ^1^H (CDCl_3_): 2.96 (dd, J = 16.9, J = 6.3, 1H), 3.07 (dd, J = 16.9, J = 5.8, 1H), 3.13 (s, 3H), 3.65–6.68 (m, 1H), 3.77 (s, 3H), 6.84 (d, J = 8.6, 1H), 6.89 (dd, J = 2.5, J = 8.6, 1H), 7.06 (d, J = 2.5, 1H). NMR ^13^C (CDCl_3_): 18.85, 26.39, 41.60, 55.60, 108.79, 111.46, 113.60, 116.94, 126.89, 137.47, 156.23, 173.60. IR, cm^−1^: 2235 (-CN), 1715 (NH-CO).

2-(1-Benzyl-5-methoxy-2-oxo-2,3-dihydro-*1H*-indol-3-yl)acetonitrile (**4h**) [31]

First, 6.50 g (0.019 mol) of cyano(*N*-benzyl-2-oxo-5-methoxyindolin-3-ylidene)acetic acid **2h**, 65 mL of ethyl acetate, 35 mL of 3N HCl and 4.00 g (0.06 mol) of zinc dust were involved in reaction. The organic phase was separated and concentrated. The dry residue was dissolved in 25 mL of 2-ethoxyethanol, refluxed for 2.5 h concentrated and filtered through silica gel pad using ethyl acetate as eluent. As a result, 3.78 g of beige solid was obtained, yield 69%; m.p. = 143 °C (m.p.^lit^ 144–145 °C [31]). NMR ^1^H (CDCl_3_): 2.76 (dd, J = 15.9, 1H), 2.96 (d, J = 15.9, 1H), 3.14 (dd, J = 16.9, J = 4.8, 2H), 3.75 (dd, J = 4.3, J = 8.6, 1H), 3.78 (s, 3H), 4.91 (d, J = 7.3, 2H), 6.67 (d, J = 8.6, 1H), 6.78 (dd, J = 2.5, J = 8.6, 1H), 7.13 (d, J = 1.7, 1H), 7.34 (m, 5H). NMR ^13^C (CDCl_3_): 19.08, 41.72, 44.03, 55.59, 109.99, 111.48, 113.71, 116.88, 124.93, 127.21, 127.75, 128.83, 135.29, 136.69, 156.27, 173.86. IR, cm^−1^: 2255 (-CN), 1710 (NH-CO).

2-(5-Bromo-1-methyl-2-oxo-2,3-dihydro-*1H*-indol-3-yl)acetonitrile (**4i**) [30]

First, 5.90 g (0.019 mol) of cyano(*N*-methyl-2-oxo-5-bromoindolin-3-ylidene)acetic acid **2i**, 30 mL of ethyl acetate, 15 mL of 3N HCl and 3.60 g (0.058 mol) of Zn were involved in reaction. The organic phase was separated and concentrated. The dry residue was dissolved in 10 mL of 2-ethoxyethanol and refluxed. As a result, 2.87 g of dark beige solid was obtained, yield 57%; m.p. = 168 °C. Lit. [31]: brown oil. NMR ^1^H (CDCl_3_): 2.73 (dd, J = 16.8, J = 8.7, 1H), 3.10 (dd, J = 16.9, J = 4.7, 1H), 3.21 (s, 3H), 3.66–3.69 (m, 1H), 6.77 (d, J = 8.3, 1H), 7.49 (d, J = 8.3, 1H), 7.60 (s, 1H). NMR ^13^C (CDCl_3_): 18.68, 26.50, 41.19, 109.99, 115.60, 116.66, 127.25, 127.40, 132.18, 143.11, 173.49. MS-EI (mz, %): 264 (M^+^, 70%), 226, 224 (100%), 208, 210 (5%), 186 (10%), 155 (10%), 146 (15%), 117 (50%), 90 (20%), 76 (20%).

2-(1,6,7-Trimethyl-2-oxo-2,3-dihydro-*1H*-indol-3-yl)acetonitrile (**4j**)

First, 3.76 g (0.015 mol) of cyano(1,6,7-trimethyl-2-oxoindolin-3-ylidene)acetic acid **2j**, 35 mL of ethyl acetate, 18 mL of 3N HCl and 2.50 g (0.04 mol) of Zn were involved in reaction. The organic phase was separated and concentrated. The dry residue was dissolved in 10 mL of 2-ethoxyethanol and refluxed. Isolation gave 1.80 g of red solid, yield 60%; m.p. = 161 °C. NMR ^1^H (DMSO-d_6_): 2.22 (s, 3H), 2.37 (s, 3H), 2.72 (dd, J = 16.9, J = 4.8, 1H), 2.96 (d, J = 15.9, 1H), 3.43 (s, 3H), 3.53 (m, 1H), 6.84 (d, J = 7.1, 1H), 7.11 (d, J = 6.9, 1H). NMR ^13^C (CDCl_3_): 13.99, 19.21, 20.83, 30.82, 40.83, 117.43, 119.60, 121.28, 124.32, 124.75, 139.11, 141.98, 176.06.

Method B:

A solution of compound **3** in mixture of ethyl acetate and 3N hydrochloric acid was vigorously stirred with excess of zinc dust for 0.5 h. Then, the organic phase was separated, washed with cold water and dried using Na_2_SO_4_. The solvent was removed in vacuum. According to this procedure, the following compounds were obtained:
2-(1-Methyl-2-oxo-2,3-dihydro-*1H*-indol-3-yl)acetonitrile (**4e**)

First, 2.000 g (11.6 mmol) of 2-(1-methyl-2-oxo-2,3-dihydro-*1H*-indol-3-ylidene)acetonitrile **3e** (was dissolved in the mixture of ethyl acetate (15.4 mL) and 3N HCl (7.7 mL) and 1.817 g of zinc dust (0.028 mol) was added. The mixture was vigorously stirred for 0.5 h. Yield of **4e**: 1.624 g, 80%. The spectral data of this compound are the same as for compound obtained using method A.

2-(1-Benzyl-2-oxo-2,3-dihydro-*1H*-indol-3-yl)acetonitrile (**4f**)

First, 2.500 g, (9.6 mmol) of 2-(1-benzyl-2-oxo-2,3-dihydro-*1H*-indol-3-ylidene)acetonitrile **3f** was dissolved in mixture of ethyl acetate (12.7 mL) and 3N HCl (6.4 mL) and 1.200 g of zinc dust (17.7 mmol) was added. Yield of **4f**: 79%, 1.985 g. The spectral data of this compound are the same as for compound obtained using method A.

2-(1-Methyl-5-methoxy-2-oxo-2,3-dihydro-*1H*-indol-3-yl)acetonitrile (**4g**)

First, 1.977 g (11.4 mmol) of 2-(1-methyl-5-methoxy-2-oxo-2,3-dihydro-*1H*-indol-3-ylidene)acetonitrile **3g** was dissolved in mixture of ethyl acetate (15 mL) and 3N HCl (7 mL) and 1.800 g of zinc dust (27.7 mmol) was added. Yield of **4g**: 72%, 1.430 g. The spectral data of this compound are the same as for the compound obtained using method A.

2-(1-Benzyl-5-methoxy-2-oxo-2,3-dihydro-*1H*-indol-3-yl)acetonitrile (**4h**)

First, 2.266 g (7.8 mmol) of 2-(1-benzyl-5-methoxy-2-oxo-2,3-dihydro-*1H*-indol-3-ylidene)acetonitrile **3h** was dissolved in mixture of ethyl acetate (10.3 mL) and 3N HCl (5.2 mL) and 1.220 g of zinc dust (18.8 mmol) was added. Yield of **4h**: 83%, 1.902 g. The spectral data of this compound are the same as for compound obtained using method A.

2-(5-Bromo-1-methyl-2-oxo-2,3-dihydro-*1H*-indol-3-yl)acetonitrile (**4i**)

First, 0.333 g (1.3 mmol) of 2-(5-bromo-1-methyl-2-oxo-2,3- dihydro-*1H*-indol-3-ylidene)acetonitrile **3i** was dissolved in mixture of ethyl acetate (20 mL) and 3N HCl (10 mL) and 1.400 g of zinc dust (25 mmol) was added. Yield of **4i**: 72%, 0.250 g. The spectral data of this compound are the same as for compound obtained using method A.

2-(1,6,7-Trimethyl-2-oxo-2,3-dihydro-*1H*-indol-3-yl)acetonitrile (**4j**)

First, 1.63 g (8 mmol) of 2-(1,6,7-trimethyl-2-oxo-2,3-dihydro-*1H*-indol-3-ylidene)acetonitrile **3j** was dissolved in mixture of ethyl acetate (20 mL) and 3N HCl (10 mL), and 1.20 g of zinc dust (20 mmol) was added. Yield of **4j**: 82%, 1.40 g. The spectral data of this compound are the same as for compound obtained using method A.

#### 3.1.4. Boc Protection/Alkylation of N-unsubstituted Compounds (**4a**,**b**)

Tert-butyl 3-(cyanomethyl)-2-oxo-2,3-dihydro-*1H*-indol-1-carboxylate (**4k**)

2-(2-Oxo-2,3-dihydro-*1H*-indol-3-yl)acetonitrile (**4a**) (1.57 g, 1.2 mmol) was dissolved in 20 mL of dry THF and solution was cooled up to −40 °C. After addition of sodium hydride (60% oil suspension) (0.05 g, 1.2 mmol) the reaction mixture was vigorously stirred for 45 min at the same temperature, then 2.06 г (12 mmol) Boc_2_O was added, and solution was stirred at −40–−20 °C for 45 min and at 25 °C for 30 min. The reaction was terminated with 1 mL of water and organic solvent was evaporated. Dry residue was washed with saturated aqueous solution of K_2_CO_3_ and cold water and extracted with CH_2_Cl_2_. Compound **4k** was obtained as orange oil after filtration through silica gel pad (yield 1.20 g, 38%). NMR ^1^H (CDCl_3_): 1.64 (s, 9H), 2.77 (dd, J = 16.9, J = 8.7, 1H), 3.11 (dd, J = 16.9, J = 4.8, 1H), 3.81 (dd, J = 8.7, J = 4.8, 1H), 7.22 (t, J = 7.6, 1H), 7.38 (t, J = 7.6, 1H), 7.51 (d, J = 7.5, 1H), 7.85 (d, J = 8.1, 1H). NMR ^13^C (CDCl_3_): 19.13, 27.87, 41.85, 85.85, 115.25, 116.75, 123.78, 124.77, 129.41, 139.91, 148.51, 172.59.

Tert-butyl 5-methoxy-3-(cyanomethyl)-2-oxo-2,3-dihydro-*1H*-indol-1-carboxylate (**4l**)

2-(5-Methoxy-2-oxo-2,3-dihydro-*1H*-indol-3-yl)acetonitrile (**4b**) (1.00 g, 5 mmol) was dissolved in 20 mL of dry THF and solution was cooled up to −40 °C. After addition of sodium hydride (60% oil suspension) (0.22 g, 5.5 mmol), the reaction mixture was vigorously stirred for 45 min at the same temperature, then 1.09 г (5 mmol) Boc_2_O was added and solution was stirred at −40–−20 °C for 45 min and at 25 °C for 30 min. The reaction was terminated with addition of 1 mL of water and organic solvent was evaporated. Dry residue was washed with saturated aqueous solution of K_2_CO_3_ and cold water and extracted with CH_2_Cl_2_. Compound **4l** was obtained as orange oil after filtration through silica gel pad (yield 0.74 g, 49%). NMR ^1^H (CDCl_3_): 1.46 (s, 9H), 2.76 (dd, J = 16.9, J = 8.2, 1H), 3.03 (dd, J = 16.9, J = 5.2, 1H), 3.73 (s, 3H, OCH_3_), 3.72–3.75 (m, 1H), 6.81 (dd, J = 9.0, J = 2.6, 1H), 6.99 (d, J = 2.6, 1H), 7.69 (d, J = 9.0, 1H). NMR ^13^C (CDCl_3_): 18.82, 27.67, 41.95, 55.28, 84.27, 109.82, 113.86, 115.98, 116.59, 125.47, 133.00, 146.40, 156.75, 172.45. IR, cm^−1^ (nujol): 2850–3000 (NH), 2260 (CN), 1550, 1650. MS-EI (70 eV), m/z: 302 (M^+^), 246 (M^+^-C(CH_3_)_3_), 202 (M^+^-Boc), 175, 158, 146, 57(100%).

1,3-Di-tert-butyl 3-cyanomethyl-2-oxo-2,3-dihydro-*1H*-indol-1,3-dicarboxylate (**4m**)

2-(2-Oxo-2,3-dihydro-*1H*-indol-3-yl)acetonitrile (**4a**) (0.960 g, 0.0052 mol) was dissolved in dry acetonitrile and 0.130 g (0.0113 mol) of DMAP was added. The reaction mixture was vigorously stirred for 5 min and 2.7 g (0.012 mol) of Boc_2_O was added. The reaction was continued for 80 h at room temperature, then a solvent was evaporated, the residue was washed with aq. NaHCO_3_ and cold water, dissolved in CH_2_Cl_2_ and dried with MgSO_4_. Orange oil (55%, 1.067 g) was obtained after column chromatography (R_f_ 0.87, eluent ethyl acetate). NMR ^1^H (CDCl_3_): 1.22 (s, 9H), 1.50 (s, 9H), 2.94 (d, J = 16.8, 1H), 3.15 (J = 16.8, 1H), 7.07–7.11 (m, 1H), 7.26–7.30 (m, 2H), 7.78 (d, J = 8.0, 1H). NMR ^13^C (CDCl_3_): 13.63, 26.90, 27.40, 56.60 (C^3^), 84.03, 84.49, 115.10, 115.23, 122.40, 124.30, 124.67, 129.87, 139.79, 147.98, 164.73, 169.74.

1,3-Di-tert-butyl 5-methoxy-3-cyanomethyl-2-oxo-2,3-dihydro-*1H*-indol-1,3-dicarboxylate (**4n**)

2-(5-Methoxy-2-oxo-2,3-dihydro-*1H*-indol-3-yl)acetonitrile (**4b**) (0.228 g, 1.13 mmol) was dissolved in dry acetonitrile and 0.028 g (0.23 mmol) of DMAP was added. The reaction mixture was vigorously stirred for 5 min, and 0.800 g (3.6 mmol) of Boc_2_O was added. The reaction was continued for 40 h at room temperature, then a solvent was evaporated, the residue was washed with aq. NaHCO_3_ and cold water, dissolved in CH_2_Cl_2_ and dried with MgSO_4_. Orange oil (10%, 0.044 g) was obtained after column chromatography (R_f_ 0.8, eluent ethyl acetate). NMR ^1^H (CDCl_3_): 1.37 (s, 9H), 1.64 (s, 9H), 3.23 (d, J = 16.7, 1H), 3.03 (J = 16.7, 1H), 3.81 (s, 3H), 6.92 (d, 1H), 6.94 (d, 1H), 7.83 (d, J = 9.6, 1H). MS-EI (70 eV), m/z: 402 (2%), 376 (2%,), 364 (2%), 320 (3%), 302 (5%), 264 (10%), 246 (7%), 220 (30%), 202 (7%), 189 (5%), 175 (25%), 158 (8%). Elemental analysis found (%): C 62.68, H 6.62, N 6.91, calculated for C_21_H_26_N_2_O_6_ (%): C 62.67, H 6.51, N 6.96.

#### 3.1.5. General Procedure of Alkylation of N-Boc/alkyl-(2-oxoindolin-3-yl)acetonitriles

A mixture *N*-Boc/alkyl-(2-oxoindolin-3yl)acetonitrile, sodium hydride (60% suspension in oil) and dry THF was stirred under argon atmosphere for 30 min. Reaction mixture was cooled with ice bath (0 °C), and excess of alkyl halide was added. After stirring at room temperature for 48 h, the solvent was evaporated and the resulting oil was washed with ice water, dichloromethane (for *N*-Boc, Bn) or chloroform (for *N*-Me) and dried with MgSO_4_. The product was obtained after column chromatography. The following compounds were obtained using this general procedure:
2-(1,3-Dimethyl-2-oxo-2,3-dihydro-*1H*-indol-3-yl)acetonitrile (**5a**) [32,33]

The compound **5a** (light-orange oil, R_f_ = 0.8 in ethyl acetate, yield 56%, 3.16 g) was obtained from 5.30 g (28 mmol) of 2-(1-methyl-2-oxoindolin-3-yl)acetonitrile **4e**, 0.75 g (30 mmol) of sodium hydride and 2 mL (30 mmol) of methyl iodide in 30 mL of dry THF. NMR ^1^H (CDCl_3_): 1.53 (s, 3H), 2.57 (d, J = 16.7, 1H), 2.86 (d, J = 16.7, 1H), 3.24 (s, 3H), 6.91 (d, J = 7.8, 1H), 7.14 (t, J = 7.8, 1H), 7.34–7.38 (m, 1H), 7.48 (d, J = 7.5, 1H). NMR ^13^C (CDCl_3_): 21.22, 24.91, 25.44, 43.93, 107.89, 115.99, 122.02, 122.08, 128.19, 130.14, 141.95, 176.44.

2-(1-Benzyl-3-methyl-2-oxo-2,3-dihydro-*1H*-indol-3-yl)acetonitrile (**5b**) [33]

The compound **5b** (R_f_ = 0.7 in hexane:ethyl acetate 8:2 (*v*/*v*), yield 57%, 1.100 g) was obtained as yellow oil from 1.780 g (6.7 mmol) of 2-(1-benzyl-2-oxoindolin-3-yl)acetonitrile **4f**, 0.177 g (75 mmol) of sodium hydride and 2 mL (30 mmol) of methyl iodide in 35 mL of dry THF. NMR ^1^H (CDCl_3_): 1.54 (s, 3H), 2.65 (d, J = 16.6, 1H), 2.86 (d, J = 16.6, 1H), 4.91 (d, J = 15.7, 1H), 4.96 (d, J = 15.8, 1H), 6.80 (d, J = 7.8, 1H), 7.06 (t, J = 7.3, 1H), 7.17–7.31 (m, 6H), 7.45 (d, J = 7.3, 1H). NMR ^13^C (CDCl_3_): 22.07, 25.55, 43.30, 44.46, 109.23, 116.27, 122.62, 122.70, 126.69, 127.26, 128.37, 128.57, 130.49, 134.98, 141.36, 177.13.

2-(3-Ethyl-1-methyl-2-oxo-2,3-dihydro-*1H*-indol-3-yl)acetonitrile (**5c**) [34]

The compound **5c** (R_f_ = 0.7 in ethyl acetate, yield 15%, 0.317 g) was obtained from 1.780 g (9.6 mmol) of 2-(1-methyl-2-oxoindolin-3-yl)acetonitrile **4e**, 0.28 g (12 mmol) of sodium hydride and 1.50 mL (12 mmol) of ethyl bromide in 15 mL of dry THF. NMR ^1^H (CDCl_3_): 1.14 (t, J = 7.0, 3H), 1.73 (m, 1H), 1.85 (m, 1H), 2.55 (d, J = 16.5, 1H), 2.78 (d, J = 16.5, 1H), 3.17 (s, 3H), 6.86 (d, J = 7.8, 1H), 7.07 (t, J = 7.8, 1H), 7.20 (t, J = 7.8, 1H), 7.35 (d, J = 7.7, 1H). NMR ^13^C (CDCl_3_): 8.38, 25.63, 26.37, 29.41, 49.57, 108.52, 116.57, 122.78, 129.18, 137.47, 143.62, 176.84.

2-(1-Methyl-3-cyanomethyl-2-oxo-2,3-dihydro-*1H*-indol-3-yl)acetonitrile (**5d**)

The compound **5d** (R_f_ = 0.75 in ethyl acetate, yield 90%, 4.030 g) was obtained from 3.860 g (18 mmol) of 2-(1-methyl-2-oxo-indolin-3-yl)acetonitrile **4e**, 0.54 g (19 mmol) of sodium hydride and 2.10 mL (36 mmol) of chloroacetonitrile in 35 mL of dry THF. NMR ^1^H (CDCl_3_): 2.76 (d, J = 16.8, 2H), 2.88 (d, J = 16.8, 2H), 3.10 (s, 3H), 6.88 (d, J = 7.8, 1H), 7.07 (t, J = 7.8, 1H), 7.31 (t, J = 7.7, 1H), 7.43 (d, J = 7.3, 1H). NMR ^13^C (CDCl_3_): 23.67, 26.03, 45.16, 108.76, 114.95, 122.82, 122.98, 125.94, 129.84, 142.62, 173.35. Elemental analysis: found for C_13_H_11_N_3_O (%): C 69.26, H 4.99, N 18.62, calculated: 69.32, H 4.92, N 18.65.

2-(1,3-Dimethyl-5-methoxy-2-oxo-2,3-dihydro-*1H*-indol-3-yl)acetonitrile (**5e**) [35]

The compound **5e** (R_f_ = 0.7 in chloroform, yield 36%, 0.24 g) was obtained from 0.63 g (2.9 mmol) of 2-(1-methyl-5-methoxy-2-oxoindolin-3-yl)acetonitrile **4g**, 0.07 g (3 mmol) of sodium hydride and 1 mL (8 mmol) of methyl iodide in 15 mL of dry THF. M.p. = 84–86 °C (m.p.^lit.^ = 84–86 °C [30]), light-brown solid. NMR ^1^H (CDCl_3_): 1.36 (s, 3H), 2.50 (d, J = 16.7, 1H), 2.72 (d, J = 16.7, 1H), 3.07 (s, 3H), 3.66 (s, 3H), 6.70 (d, J = 8.3, 1H). 6.73 (dd, J = 8.6, J = 2.3, 1H), 6.95 (d, J = 2.2, 1H). NMR ^13^C (CDCl_3_): 21.67, 25.55, 25.96, 44.73, 55.21, 108.62, 109.99, 112.71, 116.20, 131.76, 135.61, 155.83, 176.53. IR, cm^−1^ (film): 2950 (NH), 2270 (CN), 1720, 1600 (CO). MS-EI (70 eV), m/z: 230 (62%), 190 (100%), 175 (30%), 147 (30%), 111 (20%), 97 (25%), 69 (50%).

2-(1-Benzyl-3-methyl-5-methoxy-2-oxo-2,3-dihydro-*1H*-indol-3-yl)acetonitrile (**5f**) [32]

The compound **5f** (R_f_ = 0.7 in ethyl acetate, yield 45%, 0.90 g) was obtained as pale yellow oil from 1.90 g (6.5 mmol) of 2-(1-benzyl-5-methoxy-2-oxoindolin-3-yl)acetonitrile **4h**, 0.17 g (7 mmol) of sodium hydride and 2 mL (32 mmol) of methyl iodide in 25 mL of dry THF. NMR ^1^H (CDCl_3_): 1.56 (s, 3H), 2.67 (d, J = 16.6, 1H), 2.88 (d, J = 16.6, 1H), 3.74 (s, 3H), 4.88 (d, J = 15.6, 1H), 4.94 (d, J = 15.6, 1H), 6.69 (d, J = 8.6, 1H), 6.74 (dd, J = 8.6, J = 2.4, 1H), 7.09 (d, J = 2.4, 1H), 7.29 (m, 5H). NMR ^13^C (CDCl_3_): 22.35, 25.95, 43.71, 55.49, 109.99, 110.25, 113.12, 116.39, 126.92, 127.49, 128.60, 132.00, 134.82, 135.21, 156.16, 177.09.

2-(3-Ethyl-1-methyl-5-methoxy-2-oxo-2,3-dihydro-*1H*-indol-3-yl)acetonitrile (**5g**)

The compound **5g** (R_f_ = 0.8 in ethyl acetate, yield 62%, 1.16 g) was obtained from 1.67 g (7.7 mmol) of 2-(1-methyl-5-methoxy-2-oxoindolin-3-yl)acetonitrile **4g**, 0.20 g (8.3 mmol) of sodium hydride and 1.20 mL (16 mmol) of ethyl bromide in 35 mL of dry THF. NMR ^1^H (CDCl_3_): 0.59 (t, J = 7.3, 3H), 1.95–2.00 (m, 2H), 2.59 (d, J = 16.6, 1H), 2.81 (d, J = 16.6, 1H), 3.20 (s, 3H), 3.79 (s, 3H), 6.73–6.87 (m, 2H), 7.01 (s, 1H). NMR ^13^C (CDCl_3_): 8.20, 25.39, 26.21, 29.22, 49.80, 55.59, 108.74, 110.39, 113.11, 116.37, 130.06, 136.81, 156.22, 176.25. HRMS-ESI, m/z: 267.1105, calculated for C_14_H_16_N_2_O_2_Na (M+Na): 267.1104_._

2-(1-Methyl-5-methoxy-3-(cyanomethyl)-2-oxo-2,3-dihydro-*1H*-indol-3-yl)acetonitrile (**5h**)

The compound **5h** (R_f_ = 0.8 in chloroform:ethyl acetate 1:1 (*v*/*v*), yield 20%, 0.220 g) was obtained from 0.940 g (4.3 mmol) of 2-(1-methyl-5-methoxy-2-oxoindolin-3-yl)acetonitrile **4g**, 0.115 g (4.7 mmol) of sodium hydride and 1.00 mL (17 mmol) of chloroacetonitrile in 15 mL of dry THF. NMR ^1^H (CDCl_3_): 2.81 (d, J = 16.8, 2H), 2.97 (d, J = 16.8, 2H), 3.24 (s, 3H), 3.81 (s, 3H), 6.88 (d, J = 8.6, 1H), 6.95 (dd, J = 8.6, J = 2.4, 1H), 7.16 (d, J = 2.3, 1H). NMR ^13^C (CDCl_3_): 24.66, 26.90, 29.68, 46.12, 55.90, 109.98, 110.88, 114.95, 115.06, 127.53, 136.38, 156.80, 173.51. HRMS-ESI, m/z: 278.0900, calculated for C_14_H_13_N_3_O_2_Na (M+Na): 278.0900.

2-(3-Methyl-5-methoxy-2-oxo-2,3-dihydro-*1H*-indol-3-yl)acetonitrile (**5i**)

2-(5-Methoxy-2-oxo-2,3-dihydro-*1H*-indol-3-yl)acetonitrile (**4b**) (0.625 g, 3.1 mmol) was dissolved in 20 mL of dry THF and sodium hydride (60% oil suspension) (0.276 g, 6.2 mmol). The reaction mixture was vigorously stirred for 15 min, then 0.58 mL (9.3 mmol) MeI was added and solution was stirred at room temperature for 4 days. The organic solvent was evaporated, and dry residue was washed with cold water, extracted with CH_2_Cl_2_ and dried with MgSO_4_. After column chromatography separation, compounds **5i** (R_f_ = 0.7 in CHCl_3_:EtOAc = 2:1) and **5e** (R_f_ = 0.85 in CHCl_3_) were obtained.

Compound **5e** was obtained as orange solid (yield 0.124 g, 17%). The spectral data of this compound are the same as for compound obtained from **4g**.

Compound (**5i**) was obtained as bright orange solid (yield 0.163 g, 24%). M.p. 135–140 °C. NMR ^1^H (CDCl_3_): 1.53 (s, 3H), 2.65 (d, J = 16.4, 1H), 2.83 (d, J = 16.7, 1H), 3.79 (s, 3H (OMe)), 6.80 (d, J = 6.0, 1H), 6.89 (s, 1H), 7.01 (d, J = 4.7, 1H). NMR ^13^C (CDCl_3_): 22.18, 26.18, 45.84, 55.30, 110.34, 111.11, 113.82, 116.57, 132.73, 133.20, 156.26 (C-OMe), 180.06 (C=O). MS-EI (70 eV), m/z: 216 (25%), 191 (100%), 176 (67%), 148 (25%), 118 (20%), 91 (15%), 72 (30%). Elemental analysis found (%): C 66.68, H 6.62, N 12.91, calculated for C_12_H_12_N_2_O_2_ (%): C 66.65, H 5.59, N 12.96.

2-(5-Bromo-1,3-dimethyl-2-oxo-2,3-dihydro-*1H*-indol-3-yl)acetonitrile (**5j**) [36]

The compound **5j** (R_f_ = 0.63 in ethyl acetate, yield 68%, 0.87 g) was obtained from 1.23 g (4.6 mmol) of 2-(5-bromo-1-methyl-2-oxoindolin-3-yl)acetonitrile **4i,** 0.123 g (5 mmol) of sodium hydride and 1 mL (16 mmol) of methyl iodide in 25 mL of dry THF. M.p. 102 °C (m.p.^lit.^ = 101–102 °C [36]). NMR ^1^H (CDCl_3_): 1.39 (s, 3H), 2.56 (d, J = 16.7, 1H), 2.76 (d, J = 16.7, 1H), 3.11 (s, 3H), 6.72 (d, J = 8.3, 1H), 7.34 (dd, J = 8.2, 1.9, 1H), 7.46 (d, J = 1.9, 1H). NMR ^13^C (CDCl_3_): 21.68, 25.44, 26.12, 44.59, 109.85, 115.09, 115.89, 125.76, 131.50, 132.47, 141.48, 176.27.

2-(1,3,6,7-Tetramethyl-2-oxo-2,3-dihydro-*1H*-indol-3-yl)acetonitrile (**5k**)

The compound **5k** (R_f_ = 0.7 in ethyl acetate:petroleum ether 1:2 (*v*/*v*)), yield 38%, 1.10 g) was obtained from 2.70 g (12.6 mmol) of 2-(1,6,7-trimethyl-2-oxoindolin-3-yl)acetonitrile **4j**, 0.30 g (13 mmol) of sodium hydride and 2 mL (32 mmol) of methyl iodide in 15 mL of dry THF. NMR ^1^H (CDCl_3_): 1.46 (s, 3H), 2.31 (s, 3H), 2.48 (s, 3H), 2.55 (d, J = 16.7, 1H), 2.79 (d, J = 16.7, 1H), 3.54 (s, 3H), 6.93 (d, J = 7.6, 1H), 7.18 (d, J = 7.6, 1H). NMR ^13^C (CDCl_3_): 14.07, 19.12, 20.78, 26.38, 30.55, 43.99, 116.76, 120.09, 121.23, 129.59, 139.02, 140.73, 142.12, 175.61.

Tert-butyl 3-(cyanomethyl)-3-methyl-2-oxo-2,3-dihydro-*1H*-indole-1-carboxylate (**5l**)

The compound **5l** (orange oil, R_f_ = 0.8 in CH_2_Cl_2_, yield 50%, 0.215 g) was obtained from 0.41 g (1.5 mmol) of *N*-Boc-3-(cyanomethyl)-2-oxoindole **4k**, 0.04 g (1.6 mmol) of sodium hydride and 0.5 mL (80 mmol) of methyl iodide in 20 mL of dry THF. NMR ^1^H (CDCl_3_): 1.53 (s, 3H), 1.62 (s, 9H), 2.64 (d, J = 16.7, 1H), 2.84 (d, J = 16.7, 1H), 7.18–7.21 (m, 1H), 7.32–7.36 (m, 1H), 7.45 (dd, J = 7.5, J = 1.1, 1H), 7.86 (d, J = 8.2, 1H). NMR ^13^C (CDCl_3_): 23.00, 26.47, 27.73, 45.02, 84.62, 115.18, 115.95, 122.64, 124.79, 129.10, 129.55, 138.35, 148.52, 175.84. Elemental analysis found for C_16_H_18_N_2_O_3_ (%): C 67.24, H 6.40, N 9.59, calculated: 67.12, H 6.34, N 9.78.

Tert-butyl 3-(cyanomethyl)-3-ethyl-2-oxo-2,3-dihydro-*1H*-indole-1-carboxylate (**5m**)

The compound **5m** (light-orange oil, R_f_ = 0.8 in petroleum ether, yield 20%, 0.160 g) was obtained from 0.89 g (3.3 mmol) of *N*-Boc-3-(cyanomethyl)-2-oxindole **4k**, 0.078 g (3.3 mmol) of sodium hydride and 2 mL (27 mmol) of ethyl bromide in 20 mL of dry THF. NMR ^1^H (CDCl_3_): 0.68 (t, J = 7.3, 3H), 1.47 (s, 9H), 1.97–2.07 (m, 2H), 2.66 (d, J = 16.7, 1H), 2.86 (d, J = 16.7, 1H), 7.37 (t, J = 7.2, 1H), 7.43 (d, J = 7.4, 1H), 7.50 (t, J = 8.1, 1H), 7.89 (d, J = 8.2, 1H). NMR ^13^C (CDCl_3_): 8.37, 22.65, 26.09, 29.66, 56.74, 87.74, 115.22, 122.98, 124.49, 125.04, 129.38, 134.03, 139.63, 148.70, 175.65.

Tert-butyl 3-methyl-5-methoxy-3-(cyanomethyl)-2-oxo-2,3-dihydro-*1H*-indole-1-carboxylate (**5n**)

The compound **5n** (red oil, R_f_ = 0.89 in chloroform, yield 27%, 0.043 g) was obtained from 0.150 g (5 mmol) of *N*-Boc-3-(cyanomethyl)-5-methoxy-2-oxindole **4l**, 0.022 g (0.92 mmol) of sodium hydride and 0.048 mL (0.75 mmol) of methyl iodide in 7 mL of dry THF. NMR ^1^H (CDCl_3_): 1.61 (s, 3H), 1.67 (s, 9H), 3.02 (d, J = 16.4, 1H), 3.26 (d, J = 16.7, 1H), 3.86 (s, 3H (OMe)), 6.93 (d, J = 9.1, 1H), 7.08 (s, 1H), 7.84 (d, J = 8.9, 1H). IR, cm^−1^ (film): 2250 (CN), 1790, 1770, 1740, 1600 (C=O), 1350–1150 (broad), 1080, 1050, 1020, 890, 860. MS-EI (70 eV), m/z: 316 (36%, M^+^), 260 (15%), 216 (100%) (M^+^-Boc), 201 (11%, M^+^-CH_3_), 189 (58%), 176 (70%), 158 (22%), 146 (19%), 132 (20%), 117 (27%), 104 (19%), 85 (37%), 56 (86%). Elemental analysis: found for C_17_H_20_N_2_O_4_(%): C 64.55, H 6.42, N 8.83, calculated: 64.54, H 6.37, N 8.86.

Tert-butyl 3-ethyl-5-methoxy-3-(cyanomethyl)-2-oxo-2,3-dihydro-*1H*-indole-1-carboxylate (**5o**)

The compound **5o** (yellow oil, R_f_ = 0.7 in chloroform, yield 16%, 0.020 g) was obtained from 0.230 g (0.76 mmol) of *N*-Boc-3-(cyanomethyl)-5-methoxy-2-oxoindole **4l**, 0.034 g (0.83 mmol) of sodium hydride and 0.085 mL (1.14 mmol) of bromoethane in 7 mL of dry THF. NMR ^1^H (CDCl_3_): 1.63 (s, 9H), 2.22–2.34 (m, 2H), 3.03 (d, J = 16.9, 1H), 3.27 (d, J = 16.9, 1H), 3.86 (s, 3H), 6.76 (s, 1H), 6.93 (d, J = 9.9, 1H), 7.05 (d, J = 8.5, 1H).

Tert-butyl 5-methoxy-3,3-bis(cyanomethyl)-2-oxo-2,3-dihydro-*1H*-indole-1-carboxylate (**5p**)

The compound **5p** (R_f_ = 0.7 in chloroform:ethyl acetate 1:1 (*v*/*v*), yield 19%, 0.041 g) was obtained from 0.195 g (0.65 mmol) of *N*-Boc-3-(cyanomethyl)-5-methoxy-2-oxoindole **4l**, 0.028 g (0.71 mmol) of sodium hydride and 0.045 mL (0.71 mmol) of chloroacetonitrile in 7 mL of dry THF. NMR ^1^H (CDCl_3_): 1.67 (s, 9H), 2.71–2.91 (m, 2H), 2.98–3.12 (m, 2H), 3.82 (s, 3H), 6.69–6.73 (m, 1H), 6.86 (s, 1H), 6.90–6.94 (m, 1H). MS-EI (70 eV), m/z: 341 (2%), 257 (5%,), 241 (7%), 202 (50%), 189 (58%), 162 (100%), 148 (30%), 119 (27%), 77 (60%).

#### 3.1.6. General Procedure for Synthesis of spiroindolin-3-ylacetonitriles (**6**)

*N*-nitrosomethylurea was added portionwise to the solution of potassium hydroxide in Et_2_O/water mixture at 0–5 °C to obtain diazomethane. The ether layer was separated and dried with KOH for 20 min. The diazomethane solution was dried by filtering through anhydrous Na_2_SO_4_ pad, then added to the solid compound **3**, and reaction mixture was vigorously stirred for 12 h until color disappeared. The reaction mixture was concentrated, and white solid residue was dissolved in toluene and heated under reflux for 8 h. The toluene was evaporated, dry residue was washed with ether, and filtered and dried in air. The following compounds were obtained using this general procedure, described by us for **6a**, **6b** [19]:
5′-Bromo-2′-oxo-1′,2′-dihydrospiro[cyclopropane-1,3′-indole]-3-carbonitrile (**6c**) [37]

The diazomethane solution was obtained from 10.50 g (100 mmol) of *N*-nitrosomethylurea, 21.20 g (380 mmol) of KOH in 57 mL of water and 137 mL of diethyl ether. After filtration through anhydrous sodium sulfate pad, the diazomethane solution was added to 2.84 g (10.8 mmol) of 2-(5-bromo-2-oxo-2,3-dihydro-*1H*-indol-3-ylidene)acetonitrile (**3c**). The reaction mixture was stirred for 14 h at room temperature, then refluxed in 40 mL of toluene for 8 h. The compound **6c** was obtained as light brown powder. Yield 1.40 g (48%), m.p. 168–175 °C. NMR ^1^H (DMSO-d_6_): 2.43–2.50 (m, 2H), 2.72–2.76 (m, 1H), 6.93 * (dd, J = 8.4, J = 2.6, 1H), 7.11 (dd, J = 8.4, J = 2.6, 1H), 7.31 * (s, 1H), 7.36 (s, 1H), 7.46 * (d, J = 8.4, 1H), 7.57 (d, J = 8.4, 1H), 10.65 * (s, 1H), 11.01 (s, 1H); *—major diastereomer, major:minor = 5:2 (by ^1^H NMR). NMR ^13^C (DMSO-d_6_): 15.71, 20.65, 27.19, 111.31, 112.26, 114.26, 117.68, 124.64, 131.51, 143.86, 172.45.

(1R*,2R*)-6′,7′-Dimethyl-2’-oxo-1’,2’-dihydrospiro[cyclopropane-1,3’-indole]-3-carbonitrile (**6d**)

The diazomethane solution was obtained from 5.30 g (60 mmol) of *N*-nitrosomethylurea, 10.00 g (180 mmol) of KOH in 27 mL of water and 80 mL of diethyl ether. After filtration through anhydrous sodium sulfate pad, the diazomethane solution was added to 1.05 g (5 mmol) of 2-(6,7-dimethyl-2-oxo-2,3-dihydro-*1H*-indol-3-ylidene)acetonitrile **3d**. The reaction mixture was stirred for 12 h at room temperature, then refluxed in 25 mL of toluene for 8 h. The obtained mixture of compounds **6d** and 1,3-dimethylated compound **6j** was separated by column chromatography (eluent petroleum ether/ethyl acetate 8:1). Compound **6d** was obtained as beige powder, single diastereomer, R_f_ 0.1, yield 0.378 g (34%), m.p. 145–147 °C. NMR ^1^H (CDCl_3_): 1.87 (dd, J = 6.8, J = 5.1, 1H), 2.12 (dd, J = 9.5, J = 4.9, 1H), 2.23 (s, 3H), 2.32 (s, 3H), 2.44 (dd, J = 9.5, J = 7.0, 1H), 6.94 (s, 2H), 9.87 (s, 1H). NMR ^13^C (CDCl_3_): 12.80, 14.39, 19.49, 20.53, 32.19, 117.62, 119.49, 121.35, 123.46, 133.36, 137.34, 138.30, 140.17, 175.87. Elemental analysis: found for C_13_H_12_N_2_O (%): C 73.57, H 5.93, N 12.98, calculated (%): C 73.56, H 5.70, N 13.20. Compound **6j** was obtained as byproduct, white powder, R_f_ 0.25, yield 0.30 g (25%), m.p.145–147 °C. The spectral data of this compound were the same as for compound **6j** obtained from **3j**.

(1R*,2R*)-1′-Methyl-2’-oxo-1’,2’-dihydrospiro[cyclopropane-1,3′-indole]-3-carbonitrile (**6e**) [38]

The diazomethane solution was obtained from 7.5 g (85 mmol) of *N*-nitrosomethylurea, 15.2 g (270 mmol) of KOH in 41 mL of water and 124 mL of diethyl ether. After filtration through anhydrous sodium sulfate pad, the diazomethane solution was added to 1.50 g (8.2 mmol) of 2-(1-methyl-2-oxo-2,3-dihydro-*1H*-indol-3-ylidene)acetonitrile **3e**. The reaction mixture was stirred for 12 h at room temperature, then refluxed in 30 mL of toluene for 8 h. The compound **6e** was obtained as beige powder, single diastereoisomer after column chromatography. Yield 0.94 g (58%), m.p. 129–131 °C. NMR ^1^H (DMSO-d_6_): 1.91 (dd, J = 5.1, J = 6.8, 1H), 2.14 (dd, J = 5.1, J = 9.5, 1H), 2.45 (dd, J = 7.1, J = 9.46, 1H), 3.31 (s, 3H), 6.98 (d, J = 7.7, 1H), 7.16 (t, J = 7.5, 1H), 7.22 (d, J = 6.8, 1H), 7.40 (t, J = 7.5, 1H). NMR ^13^C (DMSO-d_6_): 14.81, 21.34, 26.88, 31.73, 108.71, 116.91, 118.88, 120.93, 120.90, 128.85, 144.14, 172.96. Elemental analysis: found for C_12_H_10_N_2_O (%): C 72.74, H 5.17, N 14.25, calculated (%): C 72.71, H 5.08, N 14.13.

(1R*,2R*)-1’-Methyl-5’-methoxy-2’-oxo-1’,2’-dihydrospiro[cyclopropane-1,3′-indol]-3-carbonitrile (**6g**) [39]

The diazomethane solution was obtained from 6.53 g (74 mmol) of *N*-nitrosomethylurea, 12.40 g (220 mmol) of KOH in 31 mL of water and 100 mL of diethyl ether. After filtration through anhydrous sodium sulfate pad, the diazomethane solution was added to 1.40 g (6.5 mmol) of 2-(5-methoxy-1-methyl-2-oxo-2,3-dihydro-*1H*-indol-3-ylidene)acetonitrile **3g**. The reaction mixture was stirred for 12 h at room temperature, then refluxed in 30 mL of toluene for 8 h. The compound **6g** was obtained as beige powder, single isomer after column chromatography, m.p. = 164–165 °C (m.p.^lit.^ = 164–165 °C [39]), yield 1.02 g (68%). NMR ^1^H (CDCl_3_): 1.88 (dd, J = 5.0, J = 6.8, 1H), 2.14 (dd, J = 4.9, J = 9.4, 1H), 2.46 (dd, J = 6.9, J = 9.4, 1H), 3.29 (s, 3H), 3.82 (s, 3H), 6.82 (d, J = 2.3, 1H), 6.87 (d, J = 8.6, 1H), 6.92 (dd, J = 2.3, J = 8.5, 1H). NMR ^13^C (CDCl_3_): 14.69, 21.22, 26.75, 30.81, 31.60, 55.81, 108.24, 109.08, 113.22, 116.93, 137.50, 156.03, 172.51. HRMS-ESI: 251.0791, calculated for C_13_H_12_N_2_O_2_Na (M+Na): 251.0791.

5′-Bromo-1-methyl-2′-oxo-1′,2′-dihydrospiro[cyclopropane-1,3′-indole]-3-carbonitrile (**6i**)

The diazomethane solution was obtained from 3.00 g (34 mmol) of N-nitrosomethylurea, 5.75 g (102 mmol) of KOH in 16 mL of water and 46 mL of diethyl ether. After filtration through anhydrous sodium sulfate pad, the diazomethane solution was added to 0.80 g (3 mmol) of 2-(5-bromo-1-methyl-2-oxo-2,3-dihydro-*1H*-indol-3-ylidene)acetonitrile **3i**. The reaction mixture was stirred for 12 h at room temperature, then refluxed in 10 mL of toluene for 8 h. The compound **6i** was obtained as mixture of two diastereomers. Yield 0.30 g (36%), white powder, m.p. 205–207 °C. NMR ^1^H (CDCl_3_): major diastereomer: 1.91 (dd, J = 7.1, J = 5.2, 1H), 2.15 (dd, J = 9.5, J = 5.1, 1H), 2.48 (dd, J = 9.5, J = 7.1, 1H), 3.28 (s, 3H), 6.84 (d, J = 8.3, 1H), 7.30 (d, J = 1.8, 1H), 7.50 (dd, J = 8.3, J = 1.9, 1H); minor diastereomer: 2.01 (dd, J = 9.2, J = 5.2, 1H), 2.20 (dd, J = 7.5, J = 5.2, 1H), 2.36 (dd, J = 9.2, J = 7.5, 1H), 3.31 (s, 3H), 6.83 (d, J = 8.1, 1H), 6.96 (d, J = 1.8, 1H), 7.46 (dd, J = 8.3, J = 1.8, 1H); major:minor = 4.5:1 (by ^1^H NMR). NMR ^13^C (CDCl_3_) major: 15.06, 21.54, 26.89, 31.41, 109.96, 115.44, 116.34, 122.24, 124.03, 126.11, 131.63, 143.12, 172.32. Elemental analysis: found for C_12_H_9_N_2_OBr (%): C 52.04, H 3.25, N 10.06, calculated (%): C 52.01, H 3.27, N 10.11.

1′,6′,7′-Trimethyl-2′-oxo-1′,2′-dihydrospiro[cyclopropane-1,3′-indole]-3-carbonitrile (**6j**)

The diazomethane solution was obtained from 2.57 g (27 mmol) of *N*-nitrosomethylurea, 4.80 g (80 mmol) of KOH in 13 mL of water and 38 mL of diethyl ether. After filtration through anhydrous sodium sulfate pad, the diazomethane solution was added to 0.56 g (2.5 mmol) of 2-(1,6,7-trimethyl-2-oxo-2,3-dihydro-*1H*-indol-3-ylidene)acetonitrile **3j**. The reaction mixture was stirred for 12 h at room temperature, then refluxed in 15 mL of toluene for 8 h. The compound **6j** was obtained as mixture of diastereomers. Yield 0.57 g (93%), dark orange oil. NMR ^1^H (CDCl_3_): major diastereomer 1.79–1.82 (m, 1H), 2.04–2.10 (m, 1H), 2.33 (s, 3H), 2.35–2.41 (m, 1H), 2.50 (s, 3H), 3.57 (s, 3H), 6.90 (d, J = 7.7, 1H), 6.95 (d, J = 7.7, 1H); minor diastereomer, 1.87–1.91 (m, 1H), 2.06–2.11 (m, 1H), 2.21–2.25 (m, 1H), 2.29 (s, 3H), 2.48 (s, 3H), 3.58 (s, 3H), 6.53 (d, J = 7.5, 1H), 6.84 (d, J = 7.5, 1H); major:minor = 3:1 (by ^1^H NMR). NMR ^13^C (DMSO-d_6_): 13.96, 15.09, 20.84, 21.26, 30.87, 38.65, 117.94, 119.70, 124.18, 129.69, 130.91, 138.62, 142.12, 173.98. Elemental analysis: found for major isomer C_14_H_14_N_2_O (%): C 74.40, H 6.36, N 12.42, calculated (%): C 74.31, H 6.24, N 12.38.

#### 3.1.7. General Procedure for Synthesis of N-methyl-substituted Compounds **6g**, **6i**,**6j**

Acetonitrile **6** was dissolved in dry THF, and then, sodium hydride (60% suspension in oil) was added. The mixture was stirred about 30 min before addition of excess MeI. The reaction mixture was stirred at room temperature for 48 h. Organic solvent was evaporated, oil residue was washed with ice-water, extracted by ethyl acetate and dried under MgSO_4_. The following compounds were obtained using this procedure:
1′-Methyl-5′-methoxy-2′-oxo-1′,2′-dihydrospiro[cyclopropane-1,3′-indole]-3-carbonitrile (**6g**)

To the solution of 5′-methoxy-2’-oxo-1’,2’-dihydrospiro[cyclopropane-1,3′-indol]-3-carbonitrile, **6b** (1.00 g, 4.7 mmol) in 40 mL of THF 0.14 g (6 mmol) of sodium hydride was added. The reaction mixture was stirred for 5 min, and 2 mL (32 mmol) of MeI was added. Compound (**6g**) was obtained as beige powder, yield 1.01 g (93%). Spectral data are the same as for compound major isomer obtained using general procedure 3.1.6.

5′-Bromo-1′-methyl-2′-oxo-1′,2′-dihydrospiro[cyclopropane-1,3′-indole]-3-carbonitrile (**6i**)

To the solution of 5’-bromo-2’-oxo-1’,2’-dihydrospiro[cyclopropane-1,3’-indole]-3-carbonitrile, **6c** (1.28 g, 4.9 mmol) in 40 mL of THF 0.13 g (5.4 mmol) of sodium hydride was added. The reaction mixture was stirred for 5 min, and 1 mL (16 mmol) of MeI was added. Compound (**6i**) was obtained as beige powder, yield 1.01 g (74%). Spectral data are the same as for compounds obtained using the general procedure in 3.1.6.

1′,6′,7′-Trimethyl-2′-oxo-1′,2′-dihydrospiro[cyclopropane-1,3′-indole]-3-carbonitrile (**6j**)

To the solution of 6′,7′-dimethyl-2’-oxo-1’,2’-dihydrospiro[cyclopropane-1,3’-indole]-3-carbonitrile, **6d** (1.80 g, 8.5 mmol) in 25 mL of THF 0.22 g (9 mmol) of sodium hydride was added. The reaction mixture was stirred for 5 min, and 2 mL (32 mmol) of MeI was added. Compound (**6j**) was obtained as dark orange oil, yield 1.20 g (62%). Spectral data are the same as for major isomer obtained using general procedure 3.1.6 from **3j**.

#### 3.1.8. General Procedure for Selective Reduction of Nitrile Group

Method A:

The acetonitrile **4** was dissolved in glacial acetic acid and hydrogenated at room temperature and atmospheric pressure in the presence of acetic anhydride. The platinum catalyst was filtered off, and the reaction mixture was evaporated to dry residue. The solid was washed with NaHCO_3_ and extracted with dichloromethane.

Method B:

The acetonitrile **4** was dissolved in methanol, and then, acetic anhydride and anhydrous NiCl_2_ were added. The reaction mixture was cooled, NaBH_4_ was added slowly portionwise. Reaction mixture was vigorously stirred for 4 days, the solvent was evaporated, dry residue washed with saturated solution of K_2_CO_3_ and extracted with dichloromethane.

The following compounds were synthesized according to these procedures:
*N*-[2-(5-methoxy-2-oxo-2,3-dihydro-*1H*-indol-3-yl)ethyl]acetamide (**7a**)

Method A:

First, 1.00 g (4 mmol) of (5-methoxy-2-oxoindolin-3-yl)acetonitrile **4b** was hydrogenated under vigorous stirring in 15 mL of AcOH in presence of 0.5 mL Ac_2_O with 50 mg (0.22 mmol) PtO_2_ until 0.222 l of H_2_ reacted (~4 h). Yield 1.00 g (80%), m.p. = 138–146 °C. NMR ^1^H (CDCl_3_): 1.95 (s, 3H), 1.98–2.09 (m, 1H), 2.17–2.24 (m, 1H), 3.41–3.51 (m, 2H), 3.77 (c, 3H, CH_3_O), 6.58 (br. s, 1H, NH), 6.72–6.75 (dd, J = 10.6, J = 6.4, 1H), 6.79–6.81 (d, J = 8.4, 1H), 6.9 (s, 1H), 9.04 (s,1H, NH). NMR ^13^C (CDCl_3_): 23.04 (CH_3_), 29.94, 36.96, 44.75, 55.76 (OCH_3_), 110.38, 111.09, 112.75, 130.59, 135.01, 155.76 (C_Ar_-OCH_3_), 170.82 (C=O), 180.47 (C=O). IR, cm^−1^: 3298–3200 (NH), 1650 broad (C=O), 1697 (C=O). MS-EI, m/z: 248 (35%), 206 (7%, M^+^-CH_3_CO), 189 (45%), 176 (100%, M^+^-CH_3_CONHCH_2_), 163 (17%), 117 (23%), 83 (38%). Elemental analysis: found for C_13_H_16_N_2_O_3_ (%): C 62.80, H 6.46, N 11.22, calculated (%): C 62.89, H 6.50, N 11.28.

Method B:

To the solution of 0.436 g (2.2 mmol) of (5-methoxy-2-oxoindolin-3-yl)acetonitrile **4b** in 10 mL of MeOH, 0.63 mL (6.7 mmol) of Ac_2_O, 0.018 g (0.46 mmol) of freshly prepared anhydrous NiCl_2_ and finally 0.44 g (11 mmol) NaBH_4_ were added at 0–10 °C. The reaction mixture was stirred at room temperature for 4 d, and twice per day, 0.4 g of NaBH_4_ was added additionally. Yield 0.20g (37%). The analytical data of obtained compound **7a** were identical to compound prepared according to method A.

*N*-[(1,3-dimethyl-5-methoxy-2-oxo-2,3-dihydro-*1H*-indol-3-yl)ethyl]acetamide (**7b**)

Method B:

To the solution of 0.124 г (0.54 mmol) 2-(1,3-dimethyl-5-methoxy-2-oxo-2,3-dihydro-*1H*-indol-3-yl)acetonitrile (**5e**), 0.0145 g (0.12 mmol) freshly prepared NiCl_2_, 0.16 mL (0.17 mmol) Ac_2_O in 10 mL MeOH 0.13 g (0.34 mmol) NaBH_4_ were added with cooling (10–15 °C). The reaction mixture was stirred at room temperature for 4 days, the additional 0.13 g (0.34 mmol) of NaBH_4_ was added each day twice per day. Yield 22% (0.033 g). M.p. = 125–130 °C.

NMR ^1^H (CDCl_3_): 1.30 (s, 3H), 1.84 (s, 3H), 1.94–2.24 (m, 2H), 3.15 (s, 3H), 3.17–3.21 (m, 2H), 3.79 (s, 3H, OCH_3_), 6.71 (d, J = 8.8, 1H), 6.74–6.82 (m, 4H). MS-EI (70 eV), m/z: 276 (5%), 245 (2%,), 218 (5%), 191 (50%), 174 (10%), 132 (15%), 118 (20%), 84 (100%), 72 (60%), 43 (80%). HRMS-ESI: 277.1546, calculated for C_15_H_21_N_2_O_3_ (M+H): 277.1547.

*N*-[2-(5-methoxy-2-oxo-2,3-dihydro-*1H*-indol-3-yl)ethyl]propanamide (**7c**)

Method B:

To the solution of 0.211 g (1.00 mmol) of (5-methoxy-2-oxoindolin-3-yl)acetonitrile **4b** in 10 mL of MeOH, 0.300 mL (1.0 mmol) of propionic anhydride, 0.027 g (0.21 mmol) of freshly prepared anhydrous NiCl_2_ and finally 0.200 g (5.3 mmol) NaBH_4_ were added at 0–10 °C. The reaction mixture was stirred at room temperature for 4 d and twice per day 0.100 g of NaBH_4_ were added additionally. Yield 0.177 g (68%).

NMR ^1^H (CDCl_3_): 1.26 (s, 3H), 2.04–2.08 (m, 3H), 2.18–2.28 (m, 2H), 3.20–3.27 (m, 1H), 3.73 (s, 3H), 6.68 (m, 2H), 6.93 (s, 1H). HRMS-ESI: 263.1389, calculated for C_14_H_19_N_2_O_3_ (M+H): 263.1390.

*N*-[2-(5-Methoxy-2-chloro-*1H*-indol-3-yl)ethyl]propanamide (**7e**)

Method B:

To the solution of 0.109 g (5.40 mmol) of (5-methoxy-2-chloroindolin-3-yl)acetonitrile [20] in 10 mL of MeOH, 0.14 mL (5.4 mmol) of propionic anhydride, 0.016 g (0.12 mmol) of freshly prepared anhydrous NiCl_2_ and finally 0.110 g (3 mmol) NaBH_4_ were added at 0–10 °C. The reaction mixture was stirred at room temperature for 4 d, and twice per day, 0.100 g of NaBH_4_ was added additionally. Yield 0.017 g (11%).

NMR ^1^H (CDCl_3_): 1.25–1.27 (m, 3H), 2.12–2.18 (m, 2H), 3.02–3.04 (m, 2H), 3.68–3.73 (m, 2H), 3.84 (s, 3H), 6.83 (d, J = 7.1, 1H), 6.99 (d, J = 7.2, 1H), 7.15 (s, 1H). HRMS-ESI: 281.1053, calculated for C_14_H_18_ClN_2_O_3_ (M+H): 281.1051.

#### 3.1.9. General Procedure for Synthesis of Melatonin Analogues **8**,**10**

The acetonitriles (**4**–**6**) were dissolved in anhydrous THF, and NaBH_4_ was added at 0 °C in argon atmosphere. The solution of I_2_ in THF was added dropwise during 2–4 h at 0–5 °C. The reaction mixture was stirred at reflux until the solution color turned to light yellow (~2 h). The solvent was evaporated; dry residue was washed with 3N HCl, extracted with dichloromethane and dried under Na_2_SO_4_. To the dry dichloromethane fraction, Ac_2_O was added and reaction mixture was stirred for 14 h at room temperature. The reaction mixture was evaporated, and to the dry residue, EtOAc was added. The organic fraction was washed with saturated solution of sodium hydrocarbonate, or diluted solution of potassium carbonate was added. (Note: avoid the water treatment stage for 5-acetamido derivatives because of its good water solubility.) After additional extraction with dichloromethane, chloroform or ethyl acetate organic fraction was washed with water, dried with Na_2_SO_4_, and solvent was evaporated. Residue was purified using column chromatography (eluent ethyl acetate:chloroform 1:1). The following compounds were obtained using this general procedure:
*N*-[2-(1,3-Dimethyl-2,3-dihydro-*1H*-indol-3-yl)ethyl]acetamide (**8a**)

The solution of 1.780 g (7.7 mmol) of iodine in 7 mL of THF was added portionwise to the mixture of 0.700 g (3.5 mmol) of 2-(1,3-dimethyl-2-oxo-2,3-dihydro-*1H*-indol-3-yl)acetonitrile **5a** and 0.600 g (16 mmol) of sodium borohydride in 30 mL of THF. After stirring with reflux for 5 h, the reaction was terminated by addition of 2 mL of acetic acid. Then, 2 mL of acetic anhydride was added to the reaction mixture and stirred for 14 h. The compound **8a** was obtained after purification with column chromatography (R_f_ 0.28 in ethyl acetate/CH_2_Cl_2_ 1:1) as light yellow oil that turned green on TLC. Yield 39% (0.360 g). ^1^H NMR (CDCl_3_): 1.24 (s, 3H), 1.64–1.79 (m, 2H), 1.76 (s, 3H), 2.66 (s, 3H), 2.91 (d, J = 8.9, 1H), 3.07–3.12 (m, 2H), 3.20 (d, J = 8.8, 1H), 6.41 (d, J = 7.8, 1H), 6.62 (t, J = 6.7, 1H), 6.90 (d, J = 6.6, 1H), 7.01 (t, J = 7.6, 1H). ^13^C (CDCl_3_): 22.68 (CH_3_), 25.53, 35.57, 35.80, 39.70, 42.32, 67.56, 107.26, 117.82, 121.78, 127.45, 136.62, 151.61, 169.74. HRMS-ESI: 255.1468, calculated for C_14_H_20_N_2_O(M+Na): 255.1468.

*N*-[2-(1-Benzyl-3-methyl-2,3-dihydro-*1H*-indol-3-yl)ethyl]acetamide (**8b**)

The solution of 1.840 g (7.2 mmol) of iodine in 10 mL of THF was added portionwise to the mixture of 1.000 g (3.6 mmol) of 2-(1-benzyl-3-methyl-2-oxo-2,3-dihydro-*1H*-indol-3-yl)acetonitrile **5b** and 0.680 g (18 mmol) of sodium borohydride in 40 mL of THF. After stirring with reflux for 10 h, the reaction was terminated by addition of 10 mL of 3N HCl. After extraction with 10 mL of methylene dichloride, 1 mL (10 mmol) of acetic anhydride was added to the organic fraction and stirred for 14 h. The compound **8b** was obtained as red brown oil after purification with column chromatography (R_f_ 0.32 in ethyl acetate). Yield 51% (0.555 g).

^1^H NMR (CDCl_3_): 1.35 (s, 3H), 1.76 (s, 3H), 1.78–1.89 (m, 2H), 3.05 (d, J = 9.1, 1H), 3.18–3.23 (m, 2H), 3.29 (d, J = 9.0, 1H), 4.13 (d, J = 14.7, 1H), 4.39 (d, J = 14.7, 1H), 5.91 (br. s, 1H), 6.57 (d, J = 7.8, 1H), 6.75 (t, J = 7.3, 1H), 7.04 (d, J = 7.3, 1H), 7.12 (t, J = 7.7, 1H), 7.28–7.37 (m, 5H). ^13^C NMR (CDCl_3_): 22.83, 26.18, 36.10, 40.32, 42.44, 53.29, 65.39, 107.22, 117.95, 122.24, 127.07, 127.72, 128.42, 136.38, 139.96, 151.35, 169.89. HRMS-ESI: 331.1781, calculated for C_20_H_25_N_2_ONa(M+Na): 331.1781.

*N*-[2-(1-Methyl-5-methoxy-3-ethyl-2,3-dihydro-*1H*-indol-3-yl)ethyl]acetamide (**8c**)

The solution of 0.620 g (2.4 mmol) of iodine in 5 mL of THF was added portionwise to the mixture of 0.300 g (12 mmol) of 2-(3-ethyl-1-methyl-5-methoxy-2-oxo-2,3-dihydro-*1H*-indol-3-yl)acetonitrile **5g** and 0.213 g (5.6 mmol) of sodium borohydride in 20 mL of THF. After stirring with reflux for 12 h, the reaction was terminated by addition of 10 mL of 3N HCl. After extraction with 10 mL of methylene dichloride, 0.8 mL (8 mmol) of acetic anhydride was added to the organic fraction and stirred for 14 h. The compound **8c** was obtained as light red oil after purification with column chromatography (R_f_ 0.21 in ethyl acetate). Yield 31% (0.105 g). ^1^H NMR (CDCl_3_): 0.8 (t, J = 7.5, 1H), 1.63–1.80 (m, 2H), 1.84 (s, 3H), 1.92–2.11 (m, 2H), 2.72 (s, 3H), 2.96–3.06 (m, 4H), 3.75 (s, 3H), 5.73 (br.s, 1H), 6.45 (d, J = 8.5, 1H), 6.59 (d, J = 2.5, 1H), 6.69 (dd, J = 8.5, J = 2.5, 1H). ^13^C NMR (CDCl_3_): 8.75, 23.11, 26.20, 31.67, 35.85, 36.16, 46.65, 55.77, 66.32, 108.31, 110.42, 112.11, 137.09, 147.86, 153.29, 169.83. HRMS-ESI: 247.1805, calculated for C_15_H_23_N_2_O(M+H): 247.1805.

*N*-{2-[3-(2-Acetamidoethyl)-1-methyl-2,3-dihydro-*1H*-indol-3-yl]ethyl}acetamide (**8d**)

The solution of 5.89 g (23 mmol) of iodine in 7 mL of THF was added portionwise to the mixture of 1.74 g (7.7 mmol) of 2-(1-methyl-3-cyanomethyl-2-oxo-2,3-dihydro-*1H*-indol-3-yl)acetonitrile **5d** and 1.36 g (69 mmol) of sodium borohydride in 30 mL of THF. After stirring with reflux for 27 h, the reaction was terminated by addition of 10 mL of 3N HCl. After extraction with 10 mL of methylene dichloride, 1 mL (10 mmol) of acetic anhydride was added to the organic fraction and stirred for 14 h. The compound **8d** was obtained as light yellow oil after purification with column chromatography (R_f_ 0.53 in ethyl acetate). Yield 66% (1.56 g).

^1^H NMR (CDCl_3_): 1.72–1.77 (m, 2H), 1.85-1.92 (m, 2H), 2.09 (s, 3H), 2.72–2.81 (m, 2H), 2.77 (s, 6H), 3.16–3.21 (m, 4H), 6.47 (d, J = 7.8, 1H), 6.67 (t, J = 7.8, 1H), 6.94 (d, J = 7.3, 1H), 7.07 (t, J = 7.3, 1H). ^13^C NMR (CDCl_3_): 22.19, 29.79, 33.21, 34.95, 35.78, 43.72, 61.28, 108.12, 115.09, 122.05, 128.18, 138.59, 151.26, 166.26. HRMS-ESI: 304.2020, calculated for C_17_H_26_N_3_O_2_ (M+H): 304.2020.

*N*-[2-(1,3-Dimethyl-5-methoxy-2,3-dihydro-*1H*-indol-3-yl)ethyl]acetamide (**8e**)

The solution of 1.87 g (7.4 mmol) of iodine in 5 mL of THF was added portionwise to the mixture of 0.85 g (3.7 mmol) of 2-(1,3-dimethyl-5-methoxy-2-oxo-2,3-dihydro-*1H*-indol-3-yl)acetonitrile **5e** and 0.70 g (18 mmol) of sodium borohydride in 30 mL of THF. After stirring with reflux for 18 h, the reaction was terminated by addition of 10 mL of 3N HCl. After extraction with 10 mL of methylene dichloride, 1 mL (10 mmol) of acetic anhydride was added to the organic fraction and stirred for 14 h. The compound **8e** was obtained as light red oil after purification with column chromatography (R_f_ 0.31 in ethyl acetate). Yield 72% (0.70 g). ^1^H NMR (CDCl_3_): 1.32 (s, 3H), 1.70–1.82 (m, 2H), 1.84 (s, 3H), 2.71 (br.s, 3H), 2.90–2.95 (m, 1H), 3.02–3.10 (m, 1H), 3.26–3.34 (m, 2H); 3.75 (br. s, 3H), 5.98 (br. s, 1H), 6.47 (d, J = 7.7, 1H), 6.62 (s, 1H), 6.68 (d, J = 7.2, 1H). ^13^C NMR (CDCl_3_): 21.49 (CH_3_), 22.39, 30.34, 35.87, 36.62, 42.58, 55.31, 68.35, 108.09, 109.32, 111.89, 138.54, 146.17, 153.20, 166.26. HRMS-ESI: 263.1754, calculated for C_15_H_23_N_2_O_2_(M+H): 263.1754.

*N*-[2-(1-Benzyl-3-methyl-5-methoxy-2,3-dihydro-*1H*-indol-3-yl)ethyl]acetamide (**8f**)

The solution of 0.762 g (3.0 mmol) of iodine in 7 mL of THF was added portionwise to the mixture of 0.45 g (1.5 mmol) of 2-(1-benzyl-3-methyl-5-methoxy-2-oxo-2,3-dihydro-*1H*-indol-3-yl)acetonitrile **5f** and 0.27 g (6.9 mmol) of sodium borohydride in 40 mL of THF. After stirring with reflux for 2 h, the reaction was terminated by addition of 3 mL of AcOH. Then, 2 mL (20 mmol) of acetic anhydride was added to the reaction mixture and stirred for 14 h. The compound **8f** was obtained as reddish oil after purification with column chromatography (R_f_ 0.52 in ethyl acetate:CH_2_Cl_2_). Yield 77% (0.39 g). ^1^H NMR (CDCl_3_): 1.29 (s, 3H), 1.68–1.83 (m, 2H), 1.71 (s, 3H), 2.93–2.90 (m, 1H), 3.05–3.12 (m, 1H), 3.18–3.26 (m, 2H), 3.73 (s, 3H), 3.78 (s, 3H), 3.97 (d, J = 14.2, 1H), 4.30 (d, J = 14.3, 1H), 6.01 (br.s, 1H), 6.47–6.51 (m, 1H), 6.61–6.68 (m, 2H), 7.21–7.34 (m, 5H). ^13^C NMR (CDCl_3_): 22.85, 26.03, 36.09, 40.25, 42.71, 54.82, 55.74, 66.18, 108.26, 109.69, 112.05, 127.10, 127.95, 128.40, 134.27, 138.00, 138.27, 145.61, 153.29, 169.86. HRMS-ESI: 361.1886, calculated for C_21_H_26_N_2_O_2_Na(M+Na): 361.1886.

*N*-[2-(1-Methyl-5-methoxy-2,3-dihydro-*1H*-indol-3-yl)ethyl]acetamide (**8g**)

The solution of 1.120 g (4.4 mmol) of iodine in 10 mL of THF was added portionwise to the mixture of 0.471 g (2.2 mmol) of nitrile **3e** and 0.390 g (10 mmol) of sodium borohydride in 20 mL of THF. After stirring with reflux for 5 h, the reaction was terminated by addition of 2 mL of acetic acid and 0.40 mL (4 mmol) of acetic anhydride was added to the reaction mixture and stirred for 14h. The compounds **8g** and **9g** were obtained as light red oils after purification with column chromatography.

Compound **8g**, yield 14% (0.148 g). ^1^H NMR (CDCl_3_): 1.85 (s, 3H), 1.89–1.98 (m, 2H), 2.69 (s, 3H), 3.10-3.19 (m, 2H), 3.35–3.40 (m, 3H), 3.63 (s, 3H), 5.25 (s, 1H), 6.58 (d, J = 8.01, 1H), 6.66 (s, 1H), 6.77 (d, J = 8.8, 1H). HRMS-ESI: 249.1598, calculated: 249.1597 for C_14_H_21_N_2_O_2_ (M+H)_._

Compound **9g**, yield 34% (0.186 g). ^1^H NMR (CDCl_3_): 1.94 (s, 3H), 2.93 (t, J = 6.7, 2H), 3.57 (q, J = 6.5, 2H), 3.74 (s, 3H), 3.87 (s, 3H), 5.59 (s, 1H), 6.92 (dd, J = 8.8, J = 2.4, 1H), 7.03 (d, J = 2.4, 1H), 7.21 (d, J = 8.8, 1H). ^13^C NMR (CDCl_3_): 23.13, 24.93, 32.56, 39.77, 55.77, 100.49, 109.87, 111.65, 112.86, 127.18, 127.85, 132.27, 153.55, 170.18. HRMS-ESI: 247.1442, calculated: 247.1441 for C_14_H_19_N_2_O_2_ (M+H)_._

*N*-[2-(1-Benzyl-5-methoxy-2,3-dihydro-*1H*-indol-3-yl)ethyl]acetamide (**8h**)

The solution of 1.770 g (7 mmol) of iodine in 10 mL of THF was added portionwise to the mixture of 1.010 g (3.5 mmol) of 2-(1-benzyl-5-methoxy-2-oxo-2,3-dihydro-*1H*-indol-3-yl)acetonitrile **4h** and 0.600 g (16 mmol) of sodium borohydride in 20 mL of THF. After stirring with reflux for 10 h, the reaction was terminated by addition of 15 mL of 3N HCl. After extraction with 10 mL of methylene dichloride, 0.5 mL (5 mmol) of acetic anhydride was added to the organic fraction and stirred for 14 h. The compound **8h** was obtained as light yellow oil after purification with column chromatography. Yield 11% (0.125 g). ^1^H NMR (CDCl_3_): 1.72–1.74 (m, 4H), 1.94–2.05 (m, 2H), 3.01–3.12 (m, 3H), 3.20-3.40 (m, 2H), 3. (s, 3H), 4.22 (d, J = 14.9, 1H), 4.28 (d, J = 14.9, 1H), 6.56 (d, J = 6.8, 1H), 6.73 (t, J = 7.3, 1H), 7.13 (m, 5H), 7.66 (d, J = 7.8, 1H). HRMS-ESI: 325.1911, calculated for C_20_H_25_N_2_O_2_ (M+H): 325.1911.

*N*-[2-(1-Methyl-2,3-dihydro-*1H*-indol-3-yl)ethyl]acetamide (**8i**)

The solution of 4.400 g (17.3 mmol) of iodine in 10 mL of THF was added portionwise to the mixture of 1.200 g (8.6 mmol) of 2-(1-methyl-2-oxo-2,3-dihydro-*1H*-indol-3-yl)acetonitrile **4e** and 1.520 g (40 mmol) of sodium borohydride in 20 mL of THF. After stirring with reflux for 10 h, the reaction was terminated by addition of 10 mL of 3N HCl. After extraction with methylene dichloride, 1.20 mL (13 mmol) of acetic anhydride was added to the organic fraction and stirred for 14 h. The compound **8i** was obtained as yellow oil after purification with column chromatography together with **9i** as byproduct. For compound **8i**: Yield 12% (0.225 g). NMR ^1^H (CDCl_3_): 1.27 (s, 3H), 1.64–1.77 (m, 2H), 1.87–2.06 (m, 3H), 3.23–3.26 (m, 5H), 3.66–3.75 (m, 2H), 7.26–7.29 (m, 1H), 7.48–7.51 (m, 1H), 7.59–7.63 (m, 1H), 7.82 (d, J = 8.1, 1H). MS: 216 (M+, 10%), 157 (80%), 144 (100%), 115 (20%), 89 (7%), 77 (10%). HRMS-ESI: 233.1647, calculated for C_14_H_21_N_2_O (M+H): 233.1648. For compound **9i**: 2.01 (s, 3H), 3.01 (t, J=6.8, 2H), 3.61-3.66 (m, 2H), 3.78 (s, 3H), 6.03 (br. s 1H), 6.96 (s, 1H), 7.13 (t, J=7.9, 1H), 7.25 (d, J=8.7, 1H), 7.33 (d, J=8.2, 1H), 7.59 (d, J=7.8, 1H).

*N*-[2-(5-Bromo-1,3-dimethyl-2,3-dihydro-*1H*-indol-3-yl)ethyl]acetamide (**8j**)

The solution of 1.710 g (6 mmol) of iodine in 5 mL of THF was added portionwise to the mixture of 0.870 g (3.1 mmol) of 2-(5-bromo-1,3-dimethyl-2-oxo-2,3-dihydro-*1H*-indol-3-yl)acetonitrile (**5j**) and 0.600 g (15 mmol) of sodium borohydride in 40 mL of THF. After stirring with reflux for 15 h, the reaction was terminated by addition of 10 mL of 3N HCl. After extraction with 10 mL of methylene dichloride, 0.80 mL (8 mmol) of acetic anhydride was added to the organic fraction and stirred for 14 h. The compound **8j** was obtained as yellow oil after purification with column chromatography (R_f_ 0.37 in ethyl acetate). Yield 21% (0.198 g). ^1^H NMR (CDCl_3_): 1.29 (s, 3H), 1.74 (s, 3H), 1.75–1.78 (m, 1H), 1.82–1.84 (m, 1H), 2.71 (s, 3H), 3.01 (d, J = 9.1, 1H), 3.11–3.17 (m, 2H), 3.33 (d, J = 9.4, 1H), 6.22 (br.s, 1H), 6.38 (d, J = 8.3, 1H), 7.02 (d, J = 1.9, 1H), 7.13 (dd, J = 8.3, J = 1.9, 1H). ^13^C NMR (CDCl_3_): 21.84, 25.71, 29.22, 29.34, 42.73, 60.46, 109.83, 121.11, 125.23, 130.38, 139.71, 149.51, 170.50. HRMS-ESI: 311.0753, calculated for C_14_H_20_N_2_OBr(M+H): 311.0753.

*N*-[2-(1,3,6,7-Tetramethyl-2,3-dihydro-*1H*-indol-3-yl)ethyl]acetamide (**8k**)

The solution of 1.81 g (7 mmol) of iodine in 7 mL of THF was added portionwise to the mixture of 0.814 g (3.6 mmol) of 2-(1,3,6,7-tetramethyl-2-oxo-2,3-dihydro-*1H*-indol-3-yl)acetonitrile **5k** and 0.81 g (21 mmol) of sodium borohydride in 30 mL of THF. After stirring with reflux for 15 h, the reaction was terminated by addition of 10 mL of 3N HCl. After extraction with 10 mL of methylene dichloride, 1 mL (10 mmol) of acetic anhydride was added to the organic fraction and stirred for 14 h. The compound **8k** was obtained as light orange oil after purification with column chromatography (R_f_ 0.36 in ethyl acetate). Yield 50% (0.47 g). ^1^H NMR (CDCl_3_): 1.27 (s, 3H), 1.70–1.77 (m, 2H), 1.84 (s, 3H), 2.20 (s, 3H), 2.21 (s, 3H), 2.89 (s, 3H), 3.02 (d, J = 10.1, 1H), 3.11–3.16 (m, 2H), 3.40 (d, J = 10.2, 1H), 6.72 (d, J = 7.7, 1H), 6.75 (d, J = 7.5, 1H). HRMS-ESI: 261.1962, calculated for C_16_H_25_N_2_O(M+H): 261.1961.

*N*-[2-(1,6,7-Trimethyl −2,3-dihydro-*1H*-indol-3-yl)ethyl]acetamide (**8l**)

The solution of 0.526 g (2 mmol) of iodine in 2 mL of THF was added portionwise to the mixture of 0.296 g (1.38 mmol) of 2-(1,6,7-trimethyl-2-oxo-2,3-dihydro-*1H*-indol-3-yl)acetonitrile **4j** and 0.157 g (4 mmol) of sodium borohydride in 10 mL of THF. After stirring with reflux for 8 h, the reaction was terminated by addition of 15 mL of 3N HCl. After extraction with 10 mL of methylene dichloride, 0.3 mL (3 mmol) of acetic anhydride was added to the organic fraction and stirred for 14 h. The compound **8l** was obtained as light-brown oil after purification with column chromatography. Yield 24% (0.082 g). The product of aromatization, indol **9l,** was also isolated after column chromatography For compound **8l**: ^1^H NMR (DMSO-d_6_): 1.62–1.67 (m, 1H), 1.77–1.83 (m, 2H), 1.80 (s, 3H), 2.33 (s, 3H), 2.45 (s, 3H), 2.65 (s, 3H), 2.85–2.89 (m, 1H), 3.03–3.07 (m, 1H), 3.24–3.29 (m, 2H), 6.48 (d, J = 7.7, 1H), 6.69 (d, J = 7.2, 1H). ^13^C NMR (DMSO-d_6_): 13.47, 19.47, 23.13, 25.44, 30.03, 36.37, 36.56, 63.24, 110.68, 116.19, 116.54, 125.71, 126.37, 129.07, 129.93, 134.17, 169.16. HRMS-ESI: 247.1806, calculated for C_15_H_23_N_2_O (M+H): 247.1805. For compound **9l**: ^1^H NMR (CDCl_3_): 1.94 (s, 3H), 2.40 (s, 3H), 2.65 (s, 3H), 2.88 (t, J = 6.8, 2H), 3.52-3.55 (m, 2H), 4.00 (s, 3H), 5.79 (br.s, 1H), 6.70 (s, 1H), 6.92 (d, J = 8.0, 1H), 7.28 (d, J = 8.1, 1H). ^13^C NMR (CDCl_3_): 14.36, 20.86, 23.14, 24.83, 37.22, 39.76, 110.55, 115.78, 119.35, 122.23, 127.44, 128.58, 130.06, 136.38, 170.04.

*N*-[2-(1-Benzyl-2,3-dihydro-*1H*-indol-3-yl)ethyl]acetamide (**8m**)

The solution of 3.200 g (14 mmol) of iodine in 10 mL of THF was added portionwise to the mixture of 1.100 g (4.2 mmol) of 2-(1-benzyl-2-oxo-2,3-dihydro-*1H*-indol-3-yl)acetonitrile **4f** and 1.110 g (29 mmol) of sodium borohydride in 20 mL of THF. After stirring with reflux for 4 h, the reaction was terminated by addition of 7 mL of acetic acid; then, 1 mL (10 mmol) of acetic anhydride was added, and the reaction was stirred for 14 h. The compounds **8m** and **9m** were obtained as light yellow oil after purification with column chromatography (R_f_ 0.27 in EtOAc/chloroform 1:1). Yield 10% (0.130 g). For compound **8m**: ^1^H NMR (CDCl_3_): 1.52–1.63 (m, 1H), 1.75 (s, 3H), 1.80–1.88 (m, 1H), 2.84–2.88 (m, 2H), 3.10–3.16 (m, 3H), 3.30–3.36 (m, 2H), 4.04 (d, J = 14.9, 1H), 4.10 (d, J = 14.9, 1H), 5.99–6.02 (br.s, 1H), 6.38 (d, J = 7.8, 1H), 6.55 (t, J = 7.3, 1H), 7.10–7.15 (m, 6H). MS-EI: 294 (M^+^, 7%), 203 (5%), 144 (30%), 130 (5%), 91 (100%), 77 (10%). HRMS-ESI: 309.1961, calculated for C_20_H_25_N_2_O (M+H): 309.1961. For compound **9m**: ^1^H NMR (CDCl_3_): 1.75 (s, 3H), 2.82 (t, J = 7.1, 2H), 3.34 (t, J = 6.8, 2H), 5.08 (s, 1H), 6.24 (br.s., 1H), 6.80 (s, 1H), 6.91–6.99 (m, 5H), 7.4 (t, J = 7.1, 1H), 7.17–7.20 (m, 3H), 7.48 (d, J = 7.8, 1H).

*N*-{1′-Methyl-1′,2′-dihydrospiro[cyclopropan-1,3′-indol]-3-ylmethyl}acetamide (**10a**)

The solution of 2.200 g (8.6 mmol) of iodine in 7 mL of THF was added portionwise to the mixture of 0.800 g (4 mmol) of 1′-methyl-2′-oxo-1′,2′-dihydrospiro[cyclopropane-1,3′-indole]-3-carbonitrile **6e** and 0.912 g (24 mmol) of sodium borohydride in 30 mL of THF. After stirring with reflux for 10 h, the reaction was terminated by addition of 10 mL of 3N HCl. After extraction with 10 mL of methylene dichloride, 0.6 mL (5.5 mmol) of acetic anhydride was added to the organic fraction and stirred for 14 h. The compound **10a** was obtained as light red oil after purification with column chromatography (R_f_ 0.28 in ethyl acetate). Yield 76% (0.710 g). ^1^H NMR (CDCl_3_) (major diastereomer): 0.86-0.97 (m, 2H), 1.21-1.29 (m, 1H), 2.04 (s, 3H), 2.60 (s, 3H), 2.84–2.90 (m, 1H), 3.04 (d, J = 8.5, 1H), 3.15 (d, J = 8.5, 1H), 3.31–3.38 (m, 1H), 6.37 (d, J = 7.8, 1H), 6.51 (t, J = 7.5, 1H), 6.62 (d, J = 7.2, 1H), 6.92 (t, J = 7.6, 1H). ^13^C NMR (CDCl_3_): 15.08, 21.28, 22.09, 35.32, 37.82, 39.68, 64.85, 106.68, 117.14, 120.33, 126.68, 129.78, 153.96, 169.85. HRMS-ESI: 231.1491, calculated for C_14_H_19_N_2_O_4_(M+H): 231.1491.

*N*-{5′-Bromo-1′-methyl-1′,2′-dihydrospiro[cyclopropan-1,3’-indol]-3-ylmethyl}acetamide (**10b**)

The solution of 0.92 g (3.6 mmol) of iodine in 5 mL of THF was added portionwise to the mixture of 0.50 g (18 mmol) of 5′-bromo-1′-methyl-2′-oxo-1′,2′-dihydrospiro[cyclopropane-1,3′-indole]-3-carbonitrile **6i** and 0.42 g (11 mmol) of sodium borohydride in 25 mL of THF. After stirring with reflux for 15 h, the reaction was terminated by addition of 10 mL of 3N HCl. After extraction with 10 mL of methylene dichloride, 0.80 mL (8 mmol) of acetic anhydride was added to the organic fraction and stirred for 14 h. The compound **10b** was obtained as light red oil after purification with column chromatography (R_f_ 0.18 in ethyl acetate). Yield 37% (0.21 g). ^1^H NMR (CDCl_3_): 1.08 (dd, J = 5.8, J = 5.8, 1H), 1.39–1.43 (m, 1H), 2.02 (s, 3H), 2.72 (s, 3H), 2.81–2.85 (m, 1H), 3.16 (d, J = 8.6, 1H), 3.33 (d, J = 8.5, 1H), 3.65–3.72 (m, 2H), 5.62 (s, 1H), 6.34 (d, J = 8.2, 1H), 6.77 (d, J = 2.0, 1H), 7.15 (dd, J = 8.2, J = 2.0, 1H). ^13^C NMR (CDCl_3_): 14.07, 20.93, 23.01, 27.75, 28.52, 35.81, 38.34, 60.30, 108.39, 109.39, 123.63, 129.96, 132.82, 153.83, 170.35. HRMS-ESI: 309.0597, calculated for C_14_H_18_N_2_OBr(M+H): 309.0597.

*N*-{1′,6′,7′-Trimethyl-1′,2′-dihydrospiro[cyclopropan-1,3′-indol]-3-ylmethyl}acetamide (**10c**)

The solution of 3.080 g (12 mmol) of iodine in 20 mL of THF was added portionwise to the mixture of 1.370 g (6 mmol) of 1′,6′,7′-trimethyl-2′-oxo-1′,2′-dihydrospiro[cyclopropane-1,3′-indole]-3-carbonitrile **6j** and 1.160 g (36 mmol) of sodium borohydride in 20 mL of THF. After stirring with reflux for 24 h, the reaction was terminated by addition of 10 mL of 3N HCl. After extraction with 10 mL of methylene dichloride, 0.90 mL (9 mmol) of acetic anhydride was added to the organic fraction and stirred for 14 h. The compound **10c** was obtained as yellow oil after purification with column chromatography (R_f_ 0.36 in ethyl acetate). Yield 35% (0.538 g). ^1^H NMR (CDCl_3_): 0.83–0.90 (m, 1H), 1.17–1.26 (m, 2H), 2.21 (s, 6H), 2.23 (s, 3H), 2.80-2.89 (m, 1H), 2.87 (s, 3H), 3.15 (d, J = 9.8, 1H), 3.39 (d, J = 9.8, 1H), 3.47–3.52 (m, 1H), 6.30 (d, J = 6.9, 1H), 6.65 (d, J = 7.1, 1H). ^13^C NMR (CDCl_3_): 13.78, 19.57, 20.23, 21.38, 22.06, 27.43, 29.71, 36.11, 41.50, 72.73, 115.53, 118.75, 121.77, 127.34, 129.07, 136.28, 142.49, 173.79. HRMS-ESI: 259.1805, calculated for C_16_H_23_N_2_O(M+H): 259.1805.

## 4. Conclusions

We designed and synthesized a number of new indole derivatives via corresponding 2-oxindoles using different reduction systems. We found that the NaBH_4_/NiCl_2_ reagent is effective for chemoselective cyano-group reduction in 2-oxindoles and in 2-chloroindoles, while BH_3_-based reagents lead to simultaneous reduction in both cyano- and amido-groups of the oxindole ring. For reduction of the cyano-group in 2-oxindoles containing two or more amido-groups, PtO_2_-catalyzed hydrogenation is more effective than boron-based reagents. The binding affinity to MT_1_/MT_2_ melatonin receptors of synthesized compounds was tested using radioligand binding assay. We found that the presence of the sp^3^ carbon at position 3 of the indole ring leads to a decrease in the melatonin receptor binding affinity of compounds of both 2-oxindoles and 2,3-dihydroindoles. The lipophilic substituents in position 2 of the melatonin ring can increase the binding affinity to melatonin receptors [40]. Thus 2-chloromelatonin remains the most active compound in the series.

## Data Availability

Not applicable.

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
