# Peer review of "Synthesis of New 2,3-Dihydroindole Derivatives and Evaluation of Their Melatonin Receptor Binding Affinity"

_molecules, 2022, doi:10.3390/molecules27217462_

Round 1

Reviewer 1 Report

The paper from Volkova et al. represents a nice work on melatonin receptor ligands introducing a novel but not drastically different scaffold active on MLT receptors. The work is well done but something can be improved.

In the references, some more recent examples can be added when the authors are taking about the several activities of MLT and its analogues, the most recent article is from 2013 and other interesting things have been developed meanwhile. I suggest to add Piomelli et al. J.Med.Chem. 2018, Volume 61, Issue 17, Pages 7902 - 7916 for a more recent hypotensive application of MLT derivatives and Spadoni et al. Chem. Res. Toxicol. 2019, Volume 32, Issue 1, Pages 100 - 112 for an innovative antioxidant application.

in Scheme 1, in the part below have been reported derivatives with OH and S that does not appear in the paper. If those have been synthesized and then excluded from the publication, should be removed from the scheme and X should mean just Cl has reported, not a generic Halogen (I can't see other halogens in position 2 in the discussion later on, just Cl).

Conclusion reports that the 2- substituted MLT derivatives are those that perform better with the receptor, recently Spadoni et al. Eur. J. Med. Chem. 2022, 243, 114762 reported a research on 2-substituted MLT derivatives ligands that could be cited to comment the result obtained by ligands of the paper and the potential presence of a lipophilic pocket that fits with substituents in position 2 reported in that paper can be somehow reinforced also by these results, I strongly suggest to use this while commenting the results obtained because it could give a complete idea of the state of the art on this topic.

SUPPORTING INFORMATION: Spectra should be present in a more uniform way, for example in 6g and 7b the peak picking is missing. Spectra should be displayed ALL in the same scale without cutting significant areas like for spectra 7b and 3j, please show ALL the spectra with the same scale.

Several compounds presents evident impurities Compounds 8e(in the aliphatic zone of the carbon), 8b (in the proton there is a big impurity at 2.05-2.10), 7b (proton, spectra not well resolved and it seems that there are impurities, the spectra seems that has been cutted in the area between 1.5 and 0, and it seems that there are other signals, impurities?), data about these compounds should be improved because most of them are not acceptable for the publication.

Author Response

Reviewer 1

The paper from Volkova et al. represents a nice work on melatonin receptor ligands introducing a novel but not drastically different scaffold active on MLT receptors. The work is well done but something can be improved.

In the references, some more recent examples can be added when the authors are taking about the several activities of MLT and its analogues, the most recent article is from 2013 and other interesting things have been developed meanwhile. I suggest to add Piomelli et al. J.Med.Chem. 2018, Volume 61, Issue 17, Pages 7902 - 7916 for a more recent hypotensive application of MLT derivatives and Spadoni et al. Chem. Res. Toxicol. 2019, Volume 32, Issue 1, Pages 100 - 112 for an innovative antioxidant application.

Answer: Thank you for these interesting references, all added (see Lines 29-32 and references 12-15)

in Scheme 1, in the part below have been reported derivatives with OH and S that does not appear in the paper. If those have been synthesized and then excluded from the publication, should be removed from the scheme and X should mean just Cl has reported, not a generic Halogen (I can't see other halogens in position 2 in the discussion later on, just Cl).

Answer: All corrected

Conclusion reports that the 2- substituted MLT derivatives are those that perform better with the receptor, recently Spadoni et al. Eur. J. Med. Chem. 2022, 243, 114762 reported a research on 2-substituted MLT derivatives ligands that could be cited to comment the result obtained by ligands of the paper and the potential presence of a lipophilic pocket that fits with substituents in position 2 reported in that paper can be somehow reinforced also by these results, I strongly suggest to use this while commenting the results obtained because it could give a complete idea of the state of the art on this topic.

Answer: Lines 1210-1213 in conclusion were corrected, information about lipophilicity and reference 40 were added.

SUPPORTING INFORMATION: Spectra should be present in a more uniform way, for example in 6g and 7b the peak picking is missing. Spectra should be displayed ALL in the same scale without cutting significant areas like for spectra 7b and 3j, please show ALL the spectra with the same scale.

Several compounds presents evident impurities Compounds 8e(in the aliphatic zone of the carbon), 8b (in the proton there is a big impurity at 2.05-2.10), 7b (proton, spectra not well resolved and it seems that there are impurities, the spectra seems that has been cutted in the area between 1.5 and 0, and it seems that there are other signals, impurities?), data about these compounds should be improved because most of them are not acceptable for the publication.

Answer: All spectra were corrected and ranged from 0 to 11 ppm (1HNMR)

Reviewer 2 Report

The article by Lozinskaya et. al. describes the design and synthesis of new 2,3-dihydroindole derivatives from corresponding polyfunctional 2-oxindoles followed by investigating their binding affinity to melatonin receptor. The target molecules were designed in a way to have the structural requirements and functional groups aiding in the binding to the melatonin receptors. Following the design, chemical synthesis strategy was applied to obtain the desired molecules in good yield. The chemistry for design of target molecules has been described in detail by the authors and synthesis was achieved following the various reduction procedures previously reported in literature. The radioligand binding assay was used to investigate the binding affinity of synthesized compounds to MT1/MT2 melatonin receptors.  

The authors provide a good introduction for the manuscript, briefly describing the purpose, synthetic design, and importance. The result and discussion section is detailed for the chemistry section and authors have provide experimental and characterization data to support their claims. The supplementary data has been provided by the authors to support the conclusions made.

However, the authors should address the following points before the manuscript can be considered publishable in Molecules-

· Line 124: ‘Spirocyclopropane derivatives 6а,b,g,j,k were synthesized in high yields (see table 2)’: There is no entry of compound ‘6j’ in the table. Please clarify?

· The molecules to have sp3-carbon at C-3 of indole ring was designed in order to increase the activity toward MT1/MT2 melatonin receptor, however, the best activity was exhibited by compound 2-chloromelatonin ‘7d’. So, it appears that aromaticity and planner structure of compound plays a crucial role in their activity. Since the compound 7d has better activity, why was compound ‘7e’ not evaluated for its melatonin receptor binding affinity for comparison and effect of additional functionalization of same molecule?

· Line 190: ‘First, the compounds 7a, 7c containing heteroatoms in position 2…’ I see not binding essay data for compound ‘7c’ in table 5 or SI.

·   Line 194: ‘2-chloromelatonin’ 7c was more active than melatonin with respect to….’ Compound 7c is has no 2-chloro group but a 2-oxindole moiety.

·  Binding assay should be performed for compound 7b and 7c too in order to understand the effect of alkyl groups on their activity.

· It would have been interesting to see the other derivatives of 2-chloromelatonin being synthesized followed by their binding studies to have more details of the steric/electronic effect of molecules on binding affinity, considering the good activity of compound 7d. This will also be beneficial in designing the next generation of such compounds. 

Supplementary Data:

1.     13C NMR spectrum of compound 5f in missing.

2.     Figure S9, S26, S28: The reference NMR signal for CDCl3 is not visible the 1H NMR spectra?

3.    Figure S32: 13C NMR spectrum not properly phased.

4. Figure S34: Chemical shift for NMR peaks is not in the spectrum.

Author Response

Reviewer 2

 Comments and Suggestions for Authors

The article by Lozinskaya et. al. describes the design and synthesis of new 2,3-dihydroindole derivatives from corresponding polyfunctional 2-oxindoles followed by investigating their binding affinity to melatonin receptor. The target molecules were designed in a way to have the structural requirements and functional groups aiding in the binding to the melatonin receptors. Following the design, chemical synthesis strategy was applied to obtain the desired molecules in good yield. The chemistry for design of target molecules has been described in detail by the authors and synthesis was achieved following the various reduction procedures previously reported in literature. The radioligand binding assay was used to investigate the binding affinity of synthesized compounds to MT1/MT2 melatonin receptors.  

The authors provide a good introduction for the manuscript, briefly describing the purpose, synthetic design, and importance. The result and discussion section is detailed for the chemistry section and authors have provide experimental and characterization data to support their claims. The supplementary data has been provided by the authors to support the conclusions made.

However, the authors should address the following points before the manuscript can be considered publishable in Molecules-

  • Line 124: ‘Spirocyclopropane derivatives 6а,b,g,j,k were synthesized in high yields (see table 2)’: There is no entry of compound ‘6j’ in the table. Please clarify?

Answer: It was our mistake in compound numeration. The line 139-140 were corrected.

  • The molecules to have sp3-carbon at C-3 of indole ring was designed in order to increase the activity toward MT1/MT2 melatonin receptor, however, the best activity was exhibited by compound 2-chloromelatonin ‘7d’. So, it appears that aromaticity and planner structure of compound plays a crucial role in their activity. Since the compound 7dhas better activity, why was compound ‘7e’ not evaluated for its melatonin receptor binding affinity for comparison and effect of additional functionalization of same molecule? Binding assay should be performed for compound 7b and 7c too in order to understand the effect of alkyl groups on their activity.

Answer: Yes, it must be done, but now we just can’t proceed radioligand binding assay due to the sanction. I hope, we will do it in the future.

  • Line 190:‘First, the compounds 7a, 7c containing heteroatoms in position 2…’ I see not binding essay data for compound ‘7c’ in table 5 or SI.

Answer: It was a mistake in compound numeration, compound 7c in text was corrected to 7d

  • Line 194:‘2-chloromelatonin’ 7c was more active than melatonin with respect to….’ Compound 7c is has no 2-chloro group but a 2-oxindole moiety.

Answer: It was a mistake in compound numeration, compound 7c in text was corrected to 7d

It would have been interesting to see the other derivatives of 2-chloromelatonin being synthesized followed by their binding studies to have more details of the steric/electronic effect of molecules on binding affinity, considering the good activity of compound 7d. This will also be beneficial in designing the next generation of such compounds. 

Answer: Yes, you are right, it would be interesting part of our further investigation

Supplementary Data:

  1. 13C NMR spectrum of compound 5f in missin

13C NMR was added

  1. Figure S9, S26, S28: The reference NMR signal for CDCl3is not visible the 1H NMR spectra?
  2. Figure S32: 13C NMR spectrum not properly phased.

Corrected

  1. Figure S34: Chemical shift for NMR peaks is not in the spectrum.

Corrected

Round 2

Reviewer 2 Report

I thank the authors for responding to the comments and modifying the manuscript when deemed necessary.